# Coverage, Not Averages: Semantic Stratification for Trustworthy Retrieval Evaluation

**Andrew Klearman** [1]  **Radu Revutchi** [1]  **Rohin Garg** [1]  **Rishav Chakravarti** [1]  **Samuel Marc Denton** [1]  **Yuan Xue** [1]

## Abstract

Retrieval quality is the primary bottleneck for accuracy and robustness in retrieval-augmented generation (RAG). Current evaluation relies on heuristically constructed query sets, which introduce a hidden intrinsic bias. We formalize retrieval evaluation as a statistical estimation problem, showing that metric reliability is fundamentally limited by the evaluation-set construction. We further introduce *semantic stratification*, which grounds evaluation in corpus structure by organizing documents into an interpretable global space of entity-based clusters and systematically generating queries for missing strata. This yields (1) formal semantic coverage guarantees across retrieval regimes and (2) interpretable visibility into retrieval failure modes. Experiments across multiple benchmarks and retrieval methods validate our framework. The results expose systematic coverage gaps, identify structural signals that explain variance in retrieval performance, and show that stratified evaluation yields more stable and transparent assessments while supporting more trustworthy decision-making than aggregate metrics.

## 1. Introduction

Retrieval-augmented generation (RAG) has emerged as a central paradigm for grounding large language models in external knowledge, enabling applications ranging from open-domain question answering to scientific search and enterprise assistants. Retrieval quality is a primary factor influencing the effectiveness of RAG systems. By mediating access to external knowledge, retrieval determines which information is available for generation and thus strongly affects downstream performance. Consequently, careful and reliable evaluation of retrieval systems is essential for understanding and improving RAG behavior.

Despite its importance, the construction of retrieval evaluation datasets remains comparatively under-investigated. Standard benchmarks rely on heuristically constructed query sets that attempt to approximate a "natural" query distribution. In offline and cold-start settings, however, such a distribution is ill-defined or unavailable.

Through an empirical analysis of public benchmarks from BEIR (Thakur et al., 2021), we show that evaluation queries often fail to adequately cover the semantic space of the underlying document corpus. Large semantic regions that appear frequently in documents are rarely or never exercised by evaluation queries. Crucially, these under-covered regions are precisely where retrieval performance is weakest. As a result, standard aggregate metrics can substantially overestimate true retrieval quality.

To address this issue, we formalize retrieval evaluation as a statistical estimation problem. We show when evaluation queries are drawn from a population that is *structurally heterogeneous* with respect to retrieval regimes, ignoring this structure leads to biased estimates and misleading confidence. These limitations echo broader concerns about benchmark reliability in NLP, where aggregate scores obscure systematic failure modes.

Motivated by this analysis, we introduce *semantic stratification*: a framework for grounding retrieval evaluation dataset construction in the structure of the document corpus. Our approach organizes documents into an interpretable global semantic structure via entity-based clustering, defines retrieval regimes along both semantic and structural dimensions, and curates evaluation queries to ensure explicit coverage guarantees. This enables unbiased performance estimation across retrieval regimes and exposes failure modes invisible to aggregate metrics.

We evaluate the proposed framework across multiple public datasets, retrieval methods, and evaluation metrics. These experiments illustrate how coverage-aware, stratified evaluation exposes retrieval behavior obscured by standard aggregate metrics and motivates more reliable comparison between retrieval systems.

---

[1]Scale AI. Correspondence to: Andrew Klearman <andrew.klearman@scale.ai>, Yuan Xue <yuan.xue@scale.ai>.

*Proceedings of the $43^{rd}$ International Conference on Machine Learning*, Seoul, South Korea. PMLR 306, 2026. Copyright 2026 by the author(s).

**Contributions.** This paper makes three contributions:

**Problem identification.** We identify a critical and underexplored issue in retrieval evaluation: evaluation datasets implicitly assume query homogeneity, while real-world query exhibits *structural heterogeneity* across retrieval regimes. We show that ignoring this heterogeneity leads to biased metrics and misleading conclusions.

**Methodological framework.** We formulate retrieval evaluation as a stratified statistical estimation problem and formalize the conditions required for reliable regime construction. Building on this analysis, we propose a semantic stratification framework for evaluation dataset construction that is coverage-aware and grounded in corpus structure.

**Empirical validation and insights.** We validate our framework across multiple benchmarks and retrieval methods. Our results expose systematic coverage gaps, identify structural signals explaining variance in retrieval performance, and show that stratified evaluation yields more stable, transparent and trustworthy decision-making than aggregate metrics.

## 2. Empirical Motivation

Using NFCorpus from BEIR (Thakur et al., 2021), a widely adopted retrieval benchmark, we show that aggregate metrics can mischaracterize the system performance. Figure 1 compares the distribution of evaluation queries against the underlying document corpus. We organize the corpus into 326 semantic clusters (details in Sec. 4) and map each query to the cluster of its ground-truth relevant document. Our analysis reveals two key observations.

First, there is a compositional mismatch between evaluation queries and the document corpus. Many high-volume semantic clusters receive little or no query coverage, appearing in the bottom-right region of Figure 1. In particular, we identify 26 clusters that together contain 17.3% of the corpus but are covered by only 1.1% of queries, including medically central topics such as *immune cells and their functions* (122 documents, 1 query). As a result, these prevalent semantic regions are effectively invisible to benchmark evaluation.

Second, retrieval performance varies systematically across semantic clusters. The color-coded mean nDCG@10 in Figure 1 shows that underrepresented clusters exhibit lower mean nDCG@10 when evaluated, but contribute little to aggregate scores due to their low query frequency.

Together, these effects induce a structural bias in benchmark evaluation with aggregate metrics dominated by overrepresented semantic regions and masking failures in prevalent but sparsely evaluated domains.

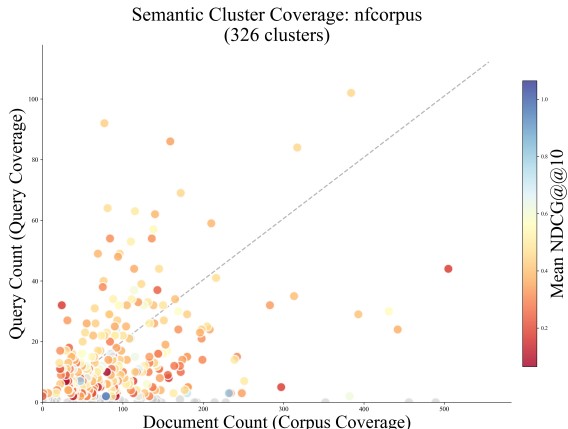

*Figure 1.* Semantic coverage in NFCorpus. Each point corresponds to a semantic cluster. The $x$-axis denotes the number of documents in the cluster, and the $y$-axis denotes the number of evaluation queries mapped to that cluster. The diagonal indicates proportional coverage. Point color encodes mean nDCG@10 for queries in the cluster; gray points indicate clusters with no query coverage.

## 3. Retrieval Evaluation as a Statistical Estimation Problem

We view retrieval evaluation as a statistical estimation problem. Let $\mathcal{D} = \{d_1, \ldots, d_N\}$ denote a document corpus and let $\mathcal{Q}$ denote the space of information-seeking queries supported by the corpus. A retrieval system is modeled as a policy $\pi : \mathcal{Q} \to \mathcal{D}^N$, which maps each query $q \in \mathcal{Q}$ to a ranked list of documents. This policy may be instantiated using different retrieval paradigms, such as dense or sparse retrieval, alternative embedding models, or document chunking strategies. Let $\phi(q; \pi)$ denote an evaluation functional, such as nDCG@k, Recall@k, or MRR@k.

Evaluation is typically performed by computing the empirical average of $\phi(q; \pi)$ over a finite set of evaluation queries. This procedure implicitly assumes that queries are sampled from the canonical distribution $\mathcal{D}$. Under this assumption, the empirical average is treated as an unbiased estimator of the system performance.

As shown in Sec. 2, this assumption is violated in realistic retrieval settings. Evaluation queries are drawn from a population that is *structuredly heterogeneous* with respect to retrieval behavior: queries fall into distinct retrieval regimes for which retrieval metrics differ systematically.

**Estimation under structured heterogeneity.** We model this structure by partitioning the query space into retrieval regimes. Let $\{\mathcal{S}_k\}_{k=1}^K$ denote a partition of $\mathcal{Q}$, where each regime corresponds to a coherent semantic or structural region of the corpus. These regimes may reflect differences in semantic intent, entity specificity, relevance dispersion, or document connectivity.

Within each regime $\mathcal{S}_k$, retrieval performance is characterized by the conditional mean $\mu_k = \mathbb{E}[\phi(q; \pi) \mid q \in \mathcal{S}_k]$ and conditional variance $\sigma_k^2 = \text{Var}[\phi(q; \pi) \mid q \in \mathcal{S}_k]$.

Let $w_k = \mathbb{P}_{q \sim Q}(q \in \mathcal{S}_k)$ denote the population mass of regime $k$. The population-level performance of $\pi$ is then

$$\mu = \sum_{k=1}^{K} w_k \mu_k. \tag{1}$$

Given an evaluation set $\mathcal{Q}_{\text{eval}}$ of size $n$, standard retrieval evaluation techniques compute the empirical mean

$$\hat{\mu}_{\text{naive}} = \frac{1}{n} \sum_{i=1}^{n} \phi(q_i; \pi). \tag{2}$$

Under the assumption that evaluation queries are i.i.d. samples from the true query distribution $Q$, $\hat{\mu}_{\text{naive}}$ is an unbiased estimator of $\mu$ with variance $\text{Var}(\phi(q; \pi))/n$. This assumption underlies common practices such as reporting single-number metrics and confidence intervals computed via bootstrapping over evaluation queries.

**Bias from coverage gaps.** Let $\hat{w}_k = \frac{n_k}{n}$ denote the empirical fraction of evaluation queries falling into regime $\mathcal{S}_k$. Conditioned on the regime composition of the evaluation set, the expected value of the naive estimator is

$$\mathbb{E}\big[\hat{\mu}_{\text{naive}} \mid \{\hat{w}_k\}_{k=1}^{K}\big] = \sum_{k=1}^{K} \hat{w}_k \mu_k. \tag{3}$$

Comparing Eq. (3) with the population mean in Eq. (1), the conditional bias of the naive estimator is

$$\text{Bias}(\hat{\mu}_{\text{naive}}) = \sum_{k=1}^{K} (\hat{w}_k - w_k) \mu_k. \tag{4}$$

In particular, if a retrieval regime $\mathcal{S}_k$ is entirely missing from the evaluation set (i.e., $\hat{w}_k = 0$ but $w_k > 0$), its contribution to Eq. (1) is systematically excluded, yielding irreducible bias regardless of evaluation set size.

**Variance mis-estimation.** Structured heterogeneity affects not only the bias of aggregate estimation, but also the uncertainty of reported metrics. Let $S(q) \in [K]$ denote the regime index of $q$. By the law of total variance,

$$\begin{aligned}\text{Var}(\phi(q; \pi)) = \,&\mathbb{E}[\text{Var}(\phi(q; \pi) \mid S(q))] \\ &+ \text{Var}(\mathbb{E}[\phi(q; \pi) \mid S(q)]),\end{aligned} \tag{5}$$

where the first term captures within-regime variability and the second term captures between-regime variability, i.e., differences in regime-level means $\{\mu_k\}_{k=1}^{K}$.

For i.i.d. queries $q_1, \ldots, q_n \sim Q$, the naive estimator satisfies $\text{Var}(\hat{\mu}_{\text{naive}}) = \text{Var}(\phi(q; \pi))/n$. When regimes with extreme values of $\mu_k$ are underrepresented or missing from $\mathcal{Q}_{\text{eval}}$, the between-regime term in Eq. (5) is systematically attenuated, yielding overly optimistic uncertainty estimates for aggregate performance.

**Regime-level performance as the primary evaluation target.** Equations (1) and (4) show that aggregate performance $\mu$ depends on the (typically unknown) regime weights $w_k$. In many retrieval settings, the deployed query distribution is ill-defined, making $\mu$ an unstable evaluation target.

In contrast, the conditional means $\mu_k = \mathbb{E}[\phi(q; \pi) \mid q \in \mathcal{S}_k]$ characterize intrinsic system behavior within coherent regions of the query space and do not depend on the global mixture weights. Whenever a regime $\mathcal{S}_k$ is represented in the evaluation set, $\mu_k$ can be estimated by the within-regime empirical mean

$$\hat{\mu}_k = \frac{1}{n_k} \sum_{i=1}^{n} \mathbb{I}\{q_i \in \mathcal{S}_k\} \phi(q_i; \pi), \tag{6}$$

This suggests that reliable evaluation should prioritize regime-level coverage, in order to accurately estimate performance profiles $\{\hat{\mu}_k\}_{k=1}^{K}$, rather than relying solely on a single aggregate metric.

**Practical conditions for regime construction.** Regime definitions should satisfy two practical requirements to support fair and actionable evaluation.

- **Method-independent regimes.** Regime construction should be independent of the particular retrieval policy being evaluated, so that the partition does not "favor" a specific method.

- **Human-interpretable descriptors.** It is desirable that each regime admits an interpretable descriptor that enables systematic diagnosis and comparison across systems.

## 4. Stratified Evaluation Dataset Construction

This section presents a practical framework for constructing evaluation datasets that satisfy the conditions introduced in Sec. 3. In particular, our goal is to ensure *within-stratum homogeneity* and *coverage awareness* by explicitly grounding evaluation set construction in the semantic and structural organization of the document corpus. We follow a three-step process: (1) corpus-level semantic structure construction, (2) query stratification, and (3) evaluation dataset curation.

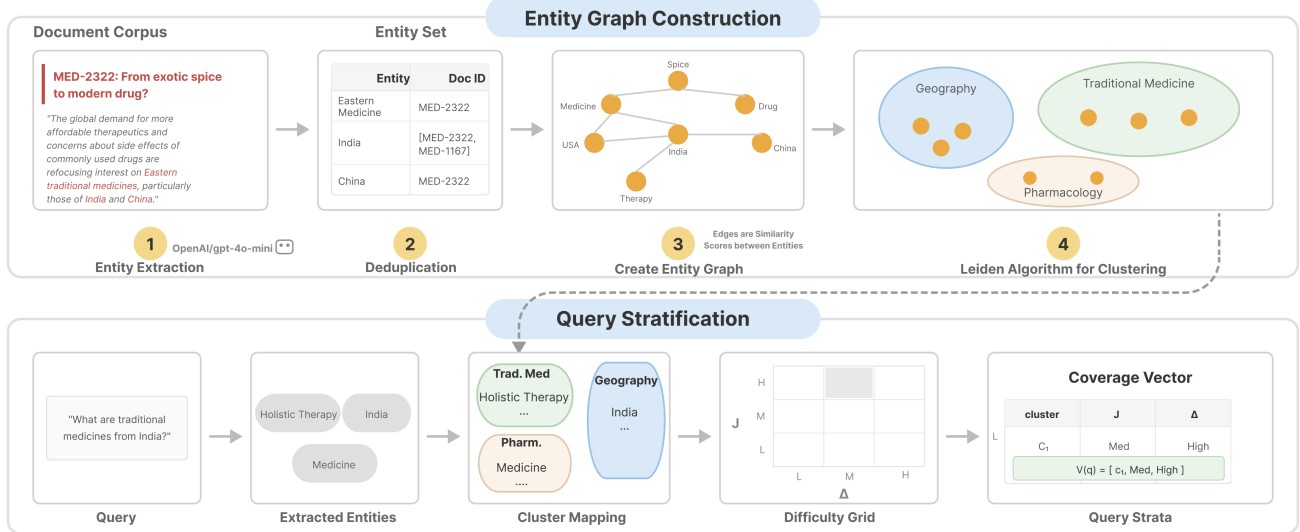

*Figure 2.* Semantic structure of a document corpus constructed via entity-based clustering. Query stratification converts query text into a set of entities. Cluster assignments, $J$, and $\Delta$ form a single query coverage vector.

## 4.1. Semantic Structure Construction

We first construct an explicit and interpretable semantic representation of the document corpus independent of any retrieval system as shown in Fig. 2

**Semantic entity extraction.** We extract *semantic entities*, which are atomic and interpretable semantic units, from each document using a LLM. Specifically, we prompt the model to identify salient entities and return a pair (name, description) that captures both surface form and semantic meaning. To improve consistency and reduce redundancy, entities are normalized and de-duplicated. Details on model selection, prompt design, example entities, and dataset-wide statistics are provided in Appendix A.1.

**Semantic graph construction.** We construct a semantic graph in which nodes correspond to normalized semantic entities. Each entity is represented by concatenating its name and description and embedding the resulting text into a fixed-dimensional vector space.[1]

Edges encode semantic relatedness. For each node, we connect a fixed number of nearest neighbors whose cosine similarity exceeds a minimum threshold, using FAISS (Johnson et al., 2019)-based approximate nearest neighbor search. Edges are undirected and weighted by cosine similarity, yielding a sparse graph that captures strong local semantic affinity. Details of the parameter choice are provided in

Appendix A.2.

**Corpus-level semantic clustering.** We apply the Leiden community detection algorithm (Traag et al., 2019) to the semantic graph to identify coherent semantic neighborhoods. The resolution parameter $\gamma$ controls cluster granularity and is tuned on a per-dataset basis to balance interpretability and coverage. As practical guidance, we recommend validating that: (1) clusters are interpretable (spot-check 10-20 cluster descriptions), (2) no single cluster dominates more than 20% of entities, and (3) fewer than 5% of entities are isolated. These checks converge quickly; our per-dataset tuning required at most 3 trials.

Each document is associated with the semantic communities containing its entities, yielding a corpus-level semantic structure. Details of the clustering procedure, resolution selection for each dataset, and sensitivity to $\gamma$ are provided in Appendix A.3 and Appendix B.2.

## 4.2. Query Stratification

Based on the corpus semantic structure, we define retrieval regimes along two categories. Together, these strata operationalize the structured heterogeneity model introduced in Sec. 3.

**Semantic strata.** Semantic strata are defined directly by the corpus-level semantic clusters. Each stratum corresponds to a distinct region of information-seeking intent supported by the corpus. This construction ensures that evaluation queries are distributed across the full semantic scope of the dataset, rather than concentrating on a narrow subset of topics.

---

[1]The embedding model is used solely to represent fine-grained semantic entities, rather than full queries, documents or chucks, and therefore does not reflect or depend on the retrieval system under evaluation. At this granularity, we observe minimal sensitivity to the specific choice of embedding model.

**Structural strata.** Retrieval difficulty varies substantially due to structural properties of the query and document relationship. We define structural strata using two complementary signals that characterize the structural complexity of retrieval regimes:

- **Relevance dispersion ($\Delta$).** Let $C(d)$ denote the set of semantic clusters associated with document $d$. We define the dispersion of a query $q$ as

$$\Delta(q) \; = \; \frac{\left| \bigcup_{d \in D(q)} C(d) \right|}{\sum_{d \in D(q)} |C(d)|}. \tag{7}$$

This quantity measures how widely the relevant documents for $q$ are spread across semantic clusters. High dispersion indicates that relevance mass is distributed across multiple semantic regions, corresponding to broad or ambiguous information needs that are typically harder for retrieval systems to rank accurately.

- **Query-document semantic alignment ($J$).** Let $C(q)$ denote the set of semantic clusters associated with entities extracted from the query, and let $C(D(q)) = \bigcup_{d \in D(q)} C(d)$ denote the clusters touched by relevant documents. We measure semantic alignment using the Jaccard similarity:

$$J(q) \; = \; \frac{|C(q) \cap C(D(q))|}{|C(q) \cup C(D(q))|}. \tag{8}$$

High alignment indicates strong correspondence between the semantic intent expressed in the query and the semantic regions covered by relevant documents, while low alignment reflects semantic mismatch that makes retrieval more error-prone.

These two signals induce a two-dimensional partitioning of queries into structural regimes with systematically different retrieval behavior. We empirically validate that the resulting $3 \times 3$ dispersion/Jaccard grid partitions queries into distinct difficulty regimes (Table 3, see also Tables 6 and 7 in Appendix B.1).

Unlike semantic strata, which are fixed by corpus structure, structural strata are defined by discretizing continuous structural signals. Given a target number of strata per signal, we compute regime boundaries from the empirical distributions of $\Delta(q)$ and $J(q)$ over an initial set of benchmark queries. These boundaries partition queries into ordered categories (e.g., low, medium, and high dispersion or alignment), yielding balanced coverage across structural regimes.

As new evaluation queries are generated or incorporated, regime boundaries may be recomputed to reflect updated dataset statistics and the desired granularity of stratification.

This adaptive construction ensures that evaluation sets maintain coverage across the full spectrum of structural regimes, preventing over-concentration in narrow subsets of queries as the evaluation set evolves.

### 4.3. Evaluation Dataset Curation

The process begins by identifying coverage gaps across entity cluster Jaccard, dispersion, and cluster coverage. Based on these gaps, the algorithm selects a generation strategy-single-cluster or multi-cluster-and samples seed documents accordingly. An LLM generates a query from the sampled document, then candidate entities are retrieved via KNN and filtered by the LLM to determine the query's semantic scope. Relevant documents are identified by pooling candidates from dense (KNN) and sparse (BM25) retrieval, verified through LLM relevance judgments. The query and its qrels are added to the evaluation set, updating coverage state. This loop repeats until the target size $|D| = N$ is reached, producing an evaluation set with guarantees on semantic coverage, structural coverage, and difficulty balance.

As the final step, we construct the evaluation dataset through an iterative query curation process that explicitly controls dataset quality. Starting from an initial query set (which may be empty), our goal is to generate additional high-quality queries to balance coverage across semantic and structural regimes. The process consists of three stages.

First, given the existing semantic and structural strata, we assess evaluation set quality using explicit **coverage criteria**. Semantic coverage measures whether semantically prevalent regions of the corpus are exercised by sufficient queries, while structural coverage measures whether queries within each semantic region span the full range of structural regimes. These metrics expose missing or underrepresented strata that are obscured by aggregate statistics.

Second, we iteratively generate candidate queries to address identified coverage gaps. At each iteration, query generation is enabled by prompting a LLM to produce multiple candidates, followed by joint LLM-based filtering and a scoring function to ensure candidate quality.

Definitions of coverage criteria, algorithmic implementation details, document sampling procedures, and query generation prompts are provided in Appendix C.

## 5. Experiments

Our experiments address three questions: (1) Do semantic strata capture meaningful structure, exhibiting internal homogeneity while revealing performance heterogeneity across retrieval regimes? (2) Do standard benchmarks exhibit coverage gaps that stratified evaluation can expose and remedy? (3) Does stratum-level analysis yield actionable

diagnostics invisible under aggregate evaluation? Across multiple datasets and retrieval systems, we answer all three questions affirmatively. Strata form coherent evaluation units with up to $3\times$ performance variation across regimes, existing benchmarks leave 40–50% of semantic regions untested, and stratified analysis reveals both retriever failure modes and corpus-level annotation issues that aggregate metrics mask entirely.

## 5.1. Experimental Setup

**Datasets and Evaluation Protocols.** We conduct experiments on three datasets from the BEIR benchmark (Thakur et al., 2021): NFCorpus (Boteva et al., 2016), FiQA (Maia et al., 2018), and Scidocs (Cohan et al., 2020). These datasets cover specialized domains in which semantic coverage is particularly important for reliable evaluation. Table 1 summarizes their key statistics.

| Dataset | Docs | Queries | Relevant |
|---|---|---|---|
| NFCorpus (Medical) | 3.6K | 323 | 38.2 |
| FiQA (Finance) | 57K | 648 | 2.6 |
| SciDocs (Scientific) | 25K | 1,000 | 4.9 |

*Table 1.* Dataset statistics for Selective BEIR corpora (Relevant indicates average relevant documents per query)

We compare two evaluation protocols: (1) **Aggregate evaluation**, where queries are sampled directly from benchmark query sets and performance is reported as the aggregate metrics with bootstrap confidence intervals; and (2) **Stratified evaluation** (ours), where queries are organized into semantic strata derived from corpus structure, ensuring explicit coverage across strata. Stratum-level metrics $\{\mu_k\}$ are computed without assuming stratum prevalence, and scalar summaries are reported using statistics across strata (e.g., medians or quantiles).

**Retrieval Systems.** We evaluate three representative retrieval paradigms: (1) **Dense** retrieval via `text-embedding-3-large`, an embedding model widely adopted in production systems (OpenAI, 2024); (2) **BM25** (Robertson & Zaragoza, 2009), a standard term-based sparse retrieval baseline; (3) **Hybrid** retrieval using Reciprocal Rank Fusion (Cormack et al., 2009) with $k = 60$ to combine dense and sparse rankings.

Across all retrieval paradigms, we observe consistent stratum-level behavior, demonstrating that our analysis is not specific to a particular retrieval paradigm.

**Metrics.** Retrieval performance is evaluated using: (1) **nDCG@10**, for both rank position and graded relevance; (2) **Recall@100**, measuring retrieval coverage; (3) **MAP@10**, averaging precision over the top-ranked results.

| Dataset | Metric | Dense | BM25 | Hybrid |
|---|---|---|---|---|
| NFCorpus | nDCG@10 | .31 / .29 | .29 / .27 | .30 / .28 |
| | Recall@100 | .29 / .27 | .25 / .22 | .29 / .26 |
| | MAP@10 | .21 / .16 | .17 / .12 | .18 / .14 |
| SciDocs | nDCG@10 | .23 / .22 | .20 / .17 | .22 / .20 |
| | Recall@100 | .30 / .28 | .27 / .25 | .29 / .27 |
| | MAP@10 | .17 / .15 | .14 / .11 | .16 / .13 |
| FiQA | nDCG@10 | .36 / .35 | .27 / .26 | .33 / .32 |
| | Recall@100 | .28 / .25 | .39 / .38 | .31 / .28 |
| | MAP@10 | .38 / .36 | .23 / .23 | .31 / .30 |

*Table 2.* Standard deviation ($\sigma_{\text{overall}}$ / $\sigma_{\text{within}}$) of retrieval metrics.

| | nDCG@10 | | | Recall@100 | | |
|---|---|---|---|---|---|---|
| | Lo $J$ | Med $J$ | Hi $J$ | Lo $J$ | Med $J$ | Hi $J$ |
| Hi $\Delta$ | .25 | .31 | .38 | .22 | .33 | .55 |
| Med $\Delta$ | .34 | .42 | .51 | .25 | .36 | .60 |
| Lo $\Delta$ | .45 | .54 | .62 | .28 | .40 | .65 |

*Table 3.* Mean nDCG@10 / Recall@100 by relevance dispersion ($\Delta$) and semantic alignment ($J$) tertiles. Dispersion dominates ranking; alignment dominates coverage.

All metrics show consistent trends across our analyses. For brevity, we report results primarily on the **NFCorpus** dataset and use **nDCG@10** as the main metric in figures. Results for additional datasets and metrics are provided in Appendix E and exhibit the same patterns.

## 5.2. Stratum Validation

We validate that our stratification framework captures meaningful structure at both the semantic and structural levels: semantic strata exhibit internal homogeneity while structural signals partition queries into distinct difficulty regimes.

**Semantic Stratum Homogeneity.** Table 2 shows that the standard deviation within strata ($\sigma_{\text{within}}$) is consistently lower than the standard deviation across the overall dataset ($\sigma_{\text{overall}}$), across all retrieval metrics, systems, and datasets.

**Performance Heterogeneity Across Strata.** Figure 3 reveals substantial performance variation across semantic strata, with different retrievers exhibiting distinct strengths and weaknesses. For example, *Nutrition Expert* is the lowest-performing stratum for all three retriever types, indicating systematic retrieval difficulty rather than model-specific failure. In contrast, *Prostate Swelling* ranks among the top strata for dense retrieval but not for sparse or hybrid methods—retriever-specific advantages masked by aggregate metrics.

**Structural Signal Effects.** The structural signals (relevance dispersion $\Delta$ and semantic alignment $J$) partition

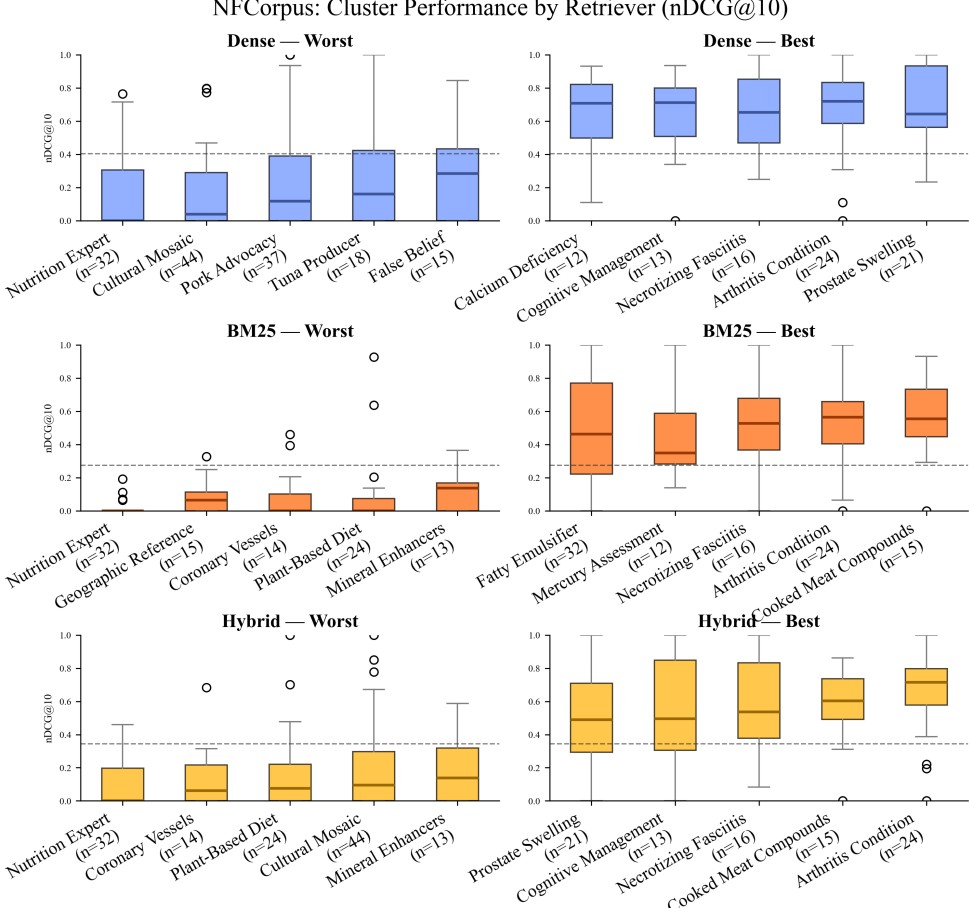

*Figure 3.* Retrieval performance (nDCG@10) across semantic clusters for different retrievers. Each box shows per-query score distribution within a cluster; dashed line indicates overall nDCG@10.

queries into difficulty regimes (Table 3). Dispersion primarily affects ranking quality (nDCG ranges from 0.25-0.62), while alignment primarily affects coverage (Recall ranges from 0.22-0.65). The diagonal from high $\Delta$/low $J$ to low $\Delta$/high $J$ spans a nearly $3\times$ performance difference, validating that structural strata represent genuinely distinct retrieval regimes. Tables 6 and 7 in Appendix B.1 confirm these patterns hold across additional datasets.

### 5.3. Coverage Analysis

We assess semantic space coverage using three metrics: **Minimum Semantic Coverage (MSC)**: the fraction of semantic clusters "touched" by at least one evaluation query. A cluster is touched if any query contains any entity belonging to that cluster. This captures whether a semantic region is tested at all.

**Sufficient Corpus Coverage (SCC)**: the fraction of documents in clusters with $\geq 5$ queries, capturing if clusters have sufficient queries for reliable per-stratum estimation.

**Zero-Query Clusters (ZQC)**: the count of semantic clusters with no evaluation queries, indicating untested semantic regions.

Table 5 reveals substantial coverage gaps: NFCorpus achieves only 51% MSC and 11% SCC. Stratified queries generated with coverage-aware sampling achieve dramatically higher coverage (90% MSC, 53% SCC) with comparable query counts.

These gaps are systematic: ZQC in NFCorpus are predominantly methodological topics (statistical methods, clinical trial designs) spanning 1,700+ documents (see Appendix D.4 for details). Note that documents in these clusters may still be retrieved for other queries, but the concepts themselves are never directly tested. A retrieval system could fail completely on such queries and the benchmark would not detect it.

We empirically confirm this in Table 4 by evaluating retrieval performance on generated queries which target two zero-query clusters. We find that the *cancer cell lines* cluster

| Zero-Query Cluster | Dense | BM25 | Hybrid |
|---|---|---|---|
| Cancer cell lines | 0.795 | **0.854** | 0.872 |
| Statistical methods | 0.365 | 0.336 | **0.483** |

*Table 4.* Retrieval performance (nDCG@10) on queries generated for zero-query clusters.

| | Aggregate | | | Stratified | | |
|---|---|---|---|---|---|---|
| | MSC | SCC | ZQC | MSC | SCC | ZQC |
| NFCorpus | 50.9% | 11% | 182 | 90.3% | 53% | 36 |
| FIQA | 60.9% | 38% | 232 | 100% | 56% | 0 |
| SciDocs | 89.2% | 48% | 69 | 99.7% | 65% | 2 |

*Table 5.* Coverage metrics for aggregate (original BEIR) vs stratified (coverage-aware generated) evaluation queries.

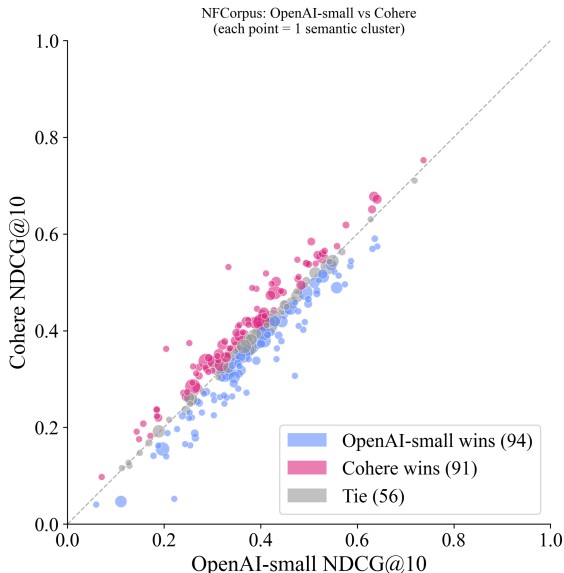

*Figure 4.* Comparing text-embedding-3-small (x-axis) and embed-english-v3.0 (y-axis). Each point represents one semantic cluster, sized by query count. Points above the diagonal indicate Cohere wins; below OpenAI wins.

achieves strong retrieval performance across all methods, yet this signal is entirely absent from the benchmark's aggregate score due to the lack of evaluation queries. In contrast, the *statistical methods* cluster represents a genuine blind spot, with substantially lower performance across all retrievers.

These two clusters further reveal distinct retriever dynamics. BM25 outperforms dense retrieval on *cancer cell lines*, while the hybrid approach provides the largest improvement on *statistical methods*, suggesting that different regions of the corpus favor different retrieval strategies. These findings reinforce our central claim that coverage is critical for reliable retrieval evaluation.

### 5.4. Interpretable Diagnostics

Stratified evaluation enables diagnosis of *why* retrieval fails within specific semantic regions, including corpus-level, *retrieval-agnostic* failure modes where success is fundamentally impossible due to limitations in corpus structure. By isolating these cases, stratified evaluation reveals failure patterns that are obscured by aggregate metrics. We present results on NFCorpus, with additional failure modes for FiQA and SciDocs reported in Appendix E

**Failure Mode 1: Semantic-Lexical Disconnect.** Several clusters reveal mismatches between query semantics and document content due to how relevance was annotated. For example, queries like "Arkansas" or "Iowa Women's Health Study" (nDCG ≈ 0.14) are judged relevant to documents discussing research *conducted in* these locations or reporting findings *from* these studies, but the names never appear in document. Similarly, researcher names and organizations are judged relevant to documents that never mention them. These queries are unanswerable from document content alone, a corpus construction issue, not a retriever limitation.

**Failure Mode 2: Register Mismatch.** In NFCorpus, health misinformation queries (nDCG = 0.14) like "Raw Food Diet Myths" use colloquial language, while relevant documents use scientific terms. This linguistic register gap makes retrieval structurally difficult regardless of method.

**Contrast: Well-Aligned Strata.** In contrast, strata with strong query-document alignment (endocrine disruptors (nDCG = 0.76), food additives (0.69), respiratory diseases (0.65)) benefit from consistent vocabulary, enabling effective semantic matching.

### 5.5. Implications on Retrieval System Decisions

In practice, retrieval evaluation guides system selection by comparing aggregate statistics. Stratified evaluation offers more intentional approaches for such decisions. We demonstrate this by comparing OpenAI's text-embedding-3-small (OpenAI, 2024) and Cohere's embed-english-v3 (Cohere, 2023) on NFCorpus (Figure 4).

When neither model dominates across all clusters, practitioners must still choose one. The conventional approach of comparing overall mean implicitly weights clusters by query count, allowing large clusters to dominate. However, this may not reflect the practitioner's actual priorities.

Figure 5 demonstrates this sensitivity through 1,000 bootstrapped evaluation sets. Using overall mean, text-embedding-3-small wins only 32.2% of samples; using macro-average (equal weight per cluster), the same model wins 63.3%. If a practitioner cares equally about all seman-

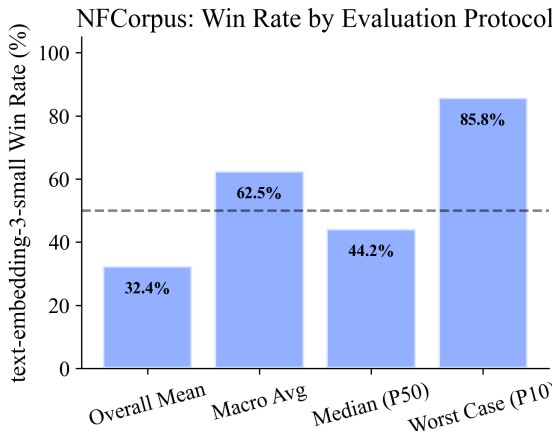

*Figure 5.* Bootstrap win rates for text-embedding-3-small vs. embed-english-v3.0 under different aggregation protocols (1,000 resamples). Overall Mean weights by query count; Macro Average weights clusters equally; Median and Worst Case capture central tendency and tail performance.

tic regions regardless of query volume, macro-averaging better reflects their intent. Stratified evaluation makes these trade-offs explicit rather than leaving them as implicit artifacts of dataset composition.

## 6. Related Work

**RAG Benchmarks**. The evaluation of RAG systems inherits methodologies from information retrieval research, most notably the Text REtrieval Conference (TREC) (Voorhees & Harman, 2005). These efforts led to unified question answering (QA) benchmarks such as BEIR, as well as foundational datasets including Natural Questions (NQ) (Kwiatkowski et al., 2019), HotpotQA (Yang et al., 2018), TriviaQA (Joshi et al., 2017), and MS MARCO (Nguyen et al., 2016) . Building on these foundations, more recent benchmarks have been developed to target specific failure modes of RAG systems, including MultiHop-RAG (Tang & Yang, 2024), CRAG (Yang et al., 2024), GlobalQA (Luo et al., 2025), Long$^2$RAG (Qi et al., 2024), MT-RAG (IBM Research, 2025), mmRAG (Xu et al., 2025), and QE-RAG (Zhang et al., 2025). However, these benchmarks largely rely on static or narrowly defined evaluation settings, leaving the limitation targeted in this paper unexamined.

**Query Taxonomies and Curation.** Prior work has proposed principled benchmark curation via query taxonomies that classify queries by task type (Teixeira de Lima et al., 2025) or operational category (Lyu et al., 2025). However, these approaches focus on query-level properties and do not address corpus-level semantic coverage gaps.

**Corpus Stratification.** Methods such as GraphRAG (Edge et al., 2025), RAPTOR (Sarthi et al., 2024), and HippoRAG (Gutierrez et al., 2024) explore corpus stratifica-

tion to improve retrieval efficiency and quality. GraphRAG is most closely related, but our approach detects relationships directly in semantic space and uses stratification for benchmark diagnosis rather than retrieval optimization.

**Synthetic Data Generation.** To reduce manual annotation costs, RAGEval (Zhu et al., 2025), Long$^2$RAG (Qi et al., 2024), and DataMorgana (Filice et al., 2025) use schema-driven pipelines to generate grounded QA data. While effective for scaling evaluation, these methods do not address systematic coverage gaps across corpus semantic regions.

**Slice-based Evaluation.** Prior work in fairness and robustness evaluation has explored slice-based analysis, where model behavior is measured across predefined subgroups or query categories that are created from poor performing inputs (Chung et al., 2019). These approaches typically define slices using query-level attributes, metadata, or task annotations. In contrast, our semantic strata are derived directly from corpus structure, enabling benchmark diagnosis over latent semantic regions without requiring manually specified taxonomies or method-specific labels.

See Appendix F for further discussion on related works.

## 7. Limitations

This work has several limitations that suggest directions for future research. First, our evaluation is conducted on a limited set of datasets and embedding models. Broader validation across more diverse domain, other languages, domains, and benchmark suites is necessary. Second, the queries used in our experiments are synthetically generated and verified by LLMs. Although this enables scalable evaluation, it may introduce biases or overlook subtle errors. Incorporating human evaluation would provide a stronger calibration and improve the reliability of the assessment. Finally, the extraction performance depends on the design of prompts, which were not exhaustively optimized. Prompt engineering could further improve extraction quality.

## 8. Conclusion

By formalizing retrieval evaluation as a statistical estimation problem under structured heterogeneity, we explain why aggregate metrics can yield biased and decision-unstable assessments. Our semantic stratification framework grounds evaluation in corpus structure, exposes previously hidden failure modes, and enables more transparent and trustworthy system comparison. We hope this work encourages a shift toward coverage-aware evaluation protocols for retrieval and RAG systems. To support reproducibility and future research, we will publicly release the stratified datasets introduced in this work.

## Impact Statement

This paper examines the reliability of retrieval evaluation benchmarks and proposes a coverage-aware, stratified evaluation framework to improve the trustworthiness of system comparison and selection. The goal of this work is to advance evaluation methodology in machine learning, particularly for retrieval and retrieval-augmented generation systems.

The primary societal impact of this work is indirect. By improving the reliability and transparency of evaluation, our framework may help practitioners make better-informed decisions when selecting retrieval systems for downstream applications, including search, question answering, and knowledge-intensive AI systems. More trustworthy evaluation may reduce the risk of deploying systems that perform well on benchmarks but fail systematically in underrepresented or high-risk information domains.

This work does not introduce new models, training procedures, or data sources, nor does it directly enable new deployment capabilities. As such, we do not anticipate immediate negative societal impacts beyond those already associated with retrieval-based machine learning systems. Ethical considerations primarily concern responsible evaluation practice, and we view improved coverage and diagnostic transparency as a positive step toward more accountable system assessment.

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

# A. Semantic Structure Construction

## A.1. Entity Extraction

```
Model: GPT-4o-mini

EXTRACTION_PROMPT = """Extract key entities from this document.

For each entity provide:
- name: the entity name (normalized, lowercase, no hyphens or special characters)
- description: a brief, context-independent description (10-20 words) that would help cluster this entity with
    semantically similar entities

Include: people, organizations, places, concepts, technical terms, methods, algorithms, systems.
Exclude: generic words, common verbs, articles.

Document:
{text}
```

**Examples of extracted entities.** Extracted entities include technical concepts (e.g., "neural network," "transformer architecture"), domain-specific terminology (e.g., "Cox regression," "quantitative easing"), named entities (e.g., "Glaxo-SmithKline," "PubMed"), and methodological terms (e.g., "randomized controlled trial").

**Entity normalization.** Entities undergo string normalization and semantic merging (e.g., "neural-networks" → "neural network") to reduce redundancy. For NFCorpus, this yields approximately 24,000 unique entities across 3,633 documents; for FiQA, over 109,000 entities from 57,000 documents.

## A.2. Graph Construction

**Entity embedding.** Each semantic entity is represented by concatenating its extracted name and description into a short text span, which is embedded using `text-embedding-3-large` text embedding model with 256 dimensions. We find that dimensionality reduction improves graph sparsity and clustering stability without materially affecting downstream analyses.

**Graph sparsification.** We construct a semantic graph by connecting each entity node to its nearest neighbors in the embedding space, using cosine similarity as the affinity measure. Approximate nearest neighbor search is performed using FAISS (Johnson et al., 2019) for scalability. For each node, we retain a fixed number of neighbors subject to a minimum similarity threshold, yielding a sparse, undirected graph with edge weights given by cosine similarity.

In all experiments, we set the number of neighbors to $k = 50$ and the similarity threshold to $0.5$. These values were selected to balance graph connectivity and sparsity, ensuring that semantic neighborhoods are well-formed while avoiding spurious long-range connections. We observe that moderate variations in these parameters do not qualitatively affect the resulting semantic communities, coverage statistics, or downstream evaluation conclusions.

## A.3. Corpus-Level Semantic Clustering

We apply the Leiden community detection algorithm (Traag et al., 2019) to the semantic graph to identify coherent semantic neighborhoods. The resolution parameter $\gamma$ controls cluster granularity and is tuned per dataset to balance interpretability and coverage ($\gamma = 25$ for NFCorpus, $\gamma = 40$ for FIQA, $\gamma = 10$ for SciDocs).

Each document is associated with all clusters containing its entities, yielding a corpus-level semantic taxonomy. Manual inspection confirms that clusters correspond to interpretable regions of the information space (e.g., diseases, methods, financial instruments).

This representation directly enables semantic coverage analysis. For NFCorpus, 42 clusters (11.3%), containing 13.5% of documents, have zero evaluation queries (systematic gaps that standard evaluation would not reveal).

# B. Query Stratification Details

## B.1. Structural Validation

**Dispersion explanation power.** On NFCorpus, relevance dispersion ($\Delta$) explains 12.3% of variance in nDCG@10 (VRR = 0.123, $p < 10^{-80}$), with low-$\Delta$ queries achieving mean nDCG of 0.56 versus 0.30 for high-$\Delta$ queries.

**Alignment explanation power.** On NFCorpus, semantic alignment ($J$) explains 23.5% of variance in Recall@100 (VRR = 0.235, $p < 10^{-100}$), with high-$J$ queries achieving mean Recall of 0.60 versus 0.25 for low-$J$ queries.

**Structural stratum partition.** We partition each signal into tertiles (Low, Medium, High), yielding a $3 \times 3$ structural grid within each semantic stratum. A complete stratum is thus identified by three components: the semantic cluster, the dispersion tertile ($\Delta$: Low, Medium, or High), and the alignment tertile ($J$: Low, Medium, or High).

**Structural signal validation.** Tables 6 and 7 extend the structural signal analysis from NFCorpus (main paper) to FiQA and SciDocs. The same pattern holds: dispersion ($\Delta$) primarily affects ranking quality (nDCG), while alignment ($J$) primarily affects coverage (Recall). The diagonal from high $\Delta$/low $J$ to low $\Delta$/high $J$ consistently represents the hardest-to-easiest progression across all datasets.

*Table 6.* FiQA: Mean nDCG@10 (left) and Recall@100 (right) by relevance dispersion ($\Delta$) and semantic alignment ($J$) tertiles (n=6,148 queries).

|  | nDCG@10 | | | Recall@100 | | |
|---|---|---|---|---|---|---|
|  | Lo $J$ | Med $J$ | Hi $J$ | Lo $J$ | Med $J$ | Hi $J$ |
| Hi $\Delta$ | 0.32 | 0.47 | 0.58 | 0.65 | 0.79 | 0.85 |
| Med $\Delta$ | 0.41 | 0.49 | 0.58 | 0.70 | 0.77 | 0.83 |
| Lo $\Delta$ | 0.46 | 0.54 | 0.59 | 0.72 | 0.80 | 0.84 |

*Table 7.* SciDocs: Mean nDCG@10 (left) and Recall@100 (right) by relevance dispersion ($\Delta$) and semantic alignment ($J$) tertiles (n=1,000 queries).

|  | nDCG@10 | | | Recall@100 | | |
|---|---|---|---|---|---|---|
|  | Lo $J$ | Med $J$ | Hi $J$ | Lo $J$ | Med $J$ | Hi $J$ |
| Hi $\Delta$ | 0.15 | 0.21 | 0.25 | 0.39 | 0.48 | 0.50 |
| Med $\Delta$ | 0.17 | 0.25 | 0.27 | 0.41 | 0.58 | 0.55 |
| Lo $\Delta$ | 0.19 | 0.28 | 0.31 | 0.45 | 0.58 | 0.62 |

In FiQA, the performance range is narrower than NFCorpus (nDCG: 0.32-0.59 vs 0.25-0.62), suggesting more uniform retrieval difficulty across structural strata. SciDocs exhibits the lowest overall performance (nDCG: 0.15-0.31), consistent with its challenging scientific citation retrieval task, but the structural patterns remain consistent: low $\Delta$ and high $J$ consistently yield better retrieval outcomes.

## B.2. Resolution Sensitivity Analysis

We conduct a resolution sensitivity analysis on NFCorpus by varying the clustering granularity parameter $\gamma \in \{10, 15, 25, 40, 60\}$.

| $\gamma$ | Clusters | $\sigma_{\text{within}}$ | $\Delta$ VRR | $J$ VRR | nDCG (min / max) |
|---|---|---|---|---|---|
| 10 | 155 | 0.310 | 0.104 | 0.219 | 0.436 / 0.632 |
| 15 | 210 | 0.305 | 0.131 | 0.228 | 0.438 / 0.643 |
| 25 | 313 | 0.306 | 0.125 | 0.237 | 0.432 / 0.739 |
| 40 | 474 | 0.305 | 0.133 | 0.286 | 0.435 / 0.856 |
| 60 | 669 | 0.304 | 0.137 | 0.326 | 0.401 / 0.856 |

*Table 8.* Sensitivity of clustering resolution $\gamma$. Core structural metrics and retrieval behavior remain stable across a $4\times$ increase in cluster count.

Across this $4\times$ range in cluster count, all core findings remain stable. Within-stratum homogeneity ($\sigma_{\text{within}}$) is consistent, and variance reduction ratios (VRR) for dispersion and Jaccard—measuring how well structural signals predict nDCG@10 and Recall@100—remain stable across resolutions.

At finer granularities, the spread in per-cluster performance increases (e.g., nDCG ranges from 0.40 to 0.86 at $\gamma = 60$), revealing increasingly differentiated retrieval behavior across semantic regions.

Qualitatively, clusters remain coherent at all resolutions; the primary difference is granularity rather than quality. For example, at $\gamma = 10$, a single cluster captures statistical modeling (285 entities), while at $\gamma = 25$ this separates into survival analysis, meta-analysis, and significance testing. At $\gamma = 60$, further refinement isolates more specific subdomains such as Cox regression and age-adjusted models.

## C. Dataset Curation

Algorithm 1 presents the coverage-aware query generation procedure.

---

**Algorithm 1** Coverage-Aware Query Generation

---

**Require:** Entity graph $G$ with clusters $\mathcal{C}$, corpus $D$ with entity mappings
**Require:** Target query count $N$, candidates per iteration $B$
  $Q \leftarrow \emptyset$ {generated queries}
  $\mathbf{n}_\sigma, \mathbf{n}_J \leftarrow \mathbf{0}$ {bucket counts for dispersion/jaccard (low, mid, high)}
  $C_{\text{covered}} \leftarrow \emptyset$ {clusters with at least one query}
  **while** $|Q| < N$ **do**
    // **Step 1: Adaptive Strategy Selection**
    $(s, m) \leftarrow \text{ChooseStrategy}(\mathbf{n}_\sigma, \mathbf{n}_J)$ {sampling strategy, entity mode}
    // **Step 2: Generate Candidate Queries**
    candidates $\leftarrow \emptyset$
    **for** $i = 1$ TO $B$ **do**
      $d_q \leftarrow \text{SampleDoc}(s)$
      $q_{\text{text}} \leftarrow \text{GenerateQuery}(d_q)$
      $D_{\text{cand}} \leftarrow \text{RetrieveRelevantDocs}(q_{\text{text}}, D, s)$
      $D_q \leftarrow \{d_q\} \cup \text{LLMVerifyRelevance}(q_{\text{text}}, D_{\text{cand}})$
      $E_q \leftarrow \text{AssignEntities}(q_{\text{text}}, m, D_q)$
      $\sigma_q \leftarrow \text{ComputeDispersion}(E_q)$
      $J_q \leftarrow \text{ComputeJaccard}(E_q, D_q)$
      candidates $\leftarrow$ candidates $\cup \{(q_{\text{text}}, D_q, \sigma_q, J_q)\}$
    **end for**
    // **Step 3: Select Best Candidate**
    $(q^*, D^*, \sigma^*, J^*) \leftarrow \arg\max_{\text{candidates}} \text{Score}(\sigma, J, \mathbf{n}_\sigma, \mathbf{n}_J, C_{\text{covered}})$
    // **Step 4: Update Coverage State**
    $Q \leftarrow Q \cup \{(q^*, D^*)\}$
    $\mathbf{n}_\sigma[\text{bucket}(\sigma^*)] \mathrel{+}= 1$
    $\mathbf{n}_J[\text{bucket}(J^*)] \mathrel{+}= 1$
    $C_{\text{covered}} \leftarrow C_{\text{covered}} \cup \text{clusters}(D^*)$
  **end while**
  **return** $Q$

---

### C.1. Prompt Specifications

**Query generation.** The following prompt is used to synthesize search queries from source documents.

```
SYSTEM:
You are an expert search query generator for information retrieval
evaluation datasets.

Your task is to generate realistic search queries that a user might
type when looking for information contained in a given document.
```

```
These queries will be used to evaluate search systems, so quality
and naturalness are critical.

CORE PRINCIPLES:
1. AUTHENTICITY: Generate queries that real users would actually
   type, not artificial constructs
2. SPECIFICITY CONTROL: Match the requested query type precisely
   (short/long, ambiguous/specific)
3. ANSWERABILITY: The query MUST be answerable by the provided
   document
4. NO LEAKAGE: Do not copy exact phrases from the document title
5. INFORMATION NEED: Express a genuine information need, not just
   keywords

WHAT TO AVOID:
- Queries that are too generic to meaningfully match the document
- Queries that require information NOT in the document
- Forced or unnatural combinations of keywords

---

USER:
Generate a search query that this document would answer.

Query Type: {query_type_description}
{query_type_instructions}
[RANDOMLY SAMPLED STYLE CONSTRAINTS - see Query Diversity Grid]
{style_constraints}

Document:
Title: {doc_title}
Content: {doc_text}

Generate ONE search query matching the specified type and style
constraints (just the query text, nothing else):
```

**Query diversity dimensions.** Style constraints are randomly sampled from this 3x2 grid to ensure diverse query characteristics.

```
QUERY DIVERSITY DIMENSIONS
==========================
For each query, we randomly sample from two orthogonal
dimensions to ensure diverse query characteristics:

DIMENSION 1: LENGTH
-------------------
- SHORT (1-2 words): "BERT embeddings", "cancer treatment"
- MEDIUM (3-4 words): "GPT-4 reasoning capabilities"
- UNRESTRICTED: Natural question length

DIMENSION 2: SPECIFICITY
------------------------
- PROPER NOUNS: Use specific named entities
  Examples: "BERT", "Pfizer", "COVID-19", "Tesla"
- GENERIC TERMS: Use category-level terms
  Examples: "language models", "vaccines", "electric vehicles"

SAMPLED STYLE CONSTRAINTS
=========================
The selected (Length, Specificity) pair injects instructions:

SHORT + PROPER NOUNS:
  LENGTH: Your query MUST be exactly 1-2 words only.
  ENTITY STYLE: Use SPECIFIC proper nouns and named entities.
  -> "BERT embeddings", "COVID-19 transmission"

SHORT + GENERIC TERMS:
  LENGTH: Your query MUST be exactly 1-2 words only.
  ENTITY STYLE: Use GENERIC category terms, NOT specific names.
  -> "language models", "cancer treatment"

MEDIUM + PROPER NOUNS:
  LENGTH: Your query MUST be exactly 3-4 words only.
  ENTITY STYLE: Use SPECIFIC proper nouns and named entities.
  -> "GPT-4 reasoning capabilities", "CRISPR Cas9 delivery"

MEDIUM + GENERIC TERMS:
  LENGTH: Your query MUST be exactly 3-4 words only.
```

```
   ENTITY STYLE: Use GENERIC category terms, NOT specific names.
   -> "deep learning applications", "gene therapy risks"

UNRESTRICTED + PROPER NOUNS:
   ENTITY STYLE: Use SPECIFIC proper nouns and named entities.
   -> "How does BERT handle contextual word representations?"

UNRESTRICTED + GENERIC TERMS:
   ENTITY STYLE: Use GENERIC category terms, NOT specific names.
   -> "How do transformer models improve understanding?"

This 3x2 sampling ensures diverse query formulations matching
real-world search patterns.
```

**Entity assignment.**    The following prompt identifies which semantic entities are relevant to a generated query.

```
SYSTEM:
You are an expert entity-to-query alignment system for information
retrieval research.

Your task is to identify which entities from a candidate list are
semantically relevant to a given search query. This mapping is used
to understand the semantic structure of queries.

WHAT COUNTS AS "RELEVANT":
1. DIRECT MENTION: The entity or a synonym appears in the query
2. IMPLICIT REFERENCE: The query is about this entity even if not
   named explicitly
   Example: "heart disease prevention" -> relevant: "cardiovascular
   disease", "heart health", "coronary artery"
3. REQUIRED KNOWLEDGE: Answering the query requires knowing about
   this entity
   Example: "CRISPR applications" -> relevant: "gene editing",
   "Cas9", "genetic engineering"
4. SEMANTIC OVERLAP: The entity's domain strongly overlaps with
   the query's information need

WHAT IS NOT RELEVANT:
- Entities that are merely topically adjacent but not needed to
  answer the query
- Entities from the same broad field but addressing different
  specific aspects
- Entities that might appear in the same document but aren't
  query-relevant

PRECISION OVER RECALL:
- It's better to miss a marginally relevant entity than to include
  an irrelevant one
- Only include entities you are confident help characterize the
  query's information need
- When uncertain, lean toward exclusion

OUTPUT: Return only the indices of relevant entities.

---

USER:
Identify which entities from the candidate list are relevant to
the search query.

An entity is RELEVANT if:
1. It appears in or is directly referenced by the query
2. It represents a key concept the query is asking about
3. Understanding this entity is necessary to answer the query
4. It's a person, organization, place, or technical term central
   to the query

An entity is NOT RELEVANT if:
- It's from the same general field but addresses a different aspect
- It might co-occur with query terms but isn't about the query topic
- It's too broad or too specific relative to the query's need

Search Query: "{query_text}"

Candidate Entities:
{entity_list}

For each candidate, ask: Would a user asking this query want/need
information about this entity? Is it central to answering the query?
```

```
Be SELECTIVE. Only include entities that genuinely characterize
the query's semantic content.

Return ONLY the entity indices that are relevant:
```

**Relevance filtering.** The following prompt identifies which retrieved documents are relevant to the query.

```
SYSTEM:
You are an expert relevance assessor for information retrieval
evaluation.

Your task is to judge whether candidate documents are RELEVANT to
a given search query. Your judgments will be used as ground truth
for evaluating search systems, so accuracy is critical.

RELEVANCE DEFINITION:
A document is RELEVANT if it contains information that would
SATISFY the user's information need expressed in the query. The
document should directly help answer or address the query.

RELEVANCE LEVELS (for your internal reasoning):
- HIGHLY RELEVANT: Document directly and completely addresses query
- RELEVANT: Document provides useful information that partially
  addresses the query
- MARGINALLY RELEVANT: Document has some topical overlap but
  limited utility
- NOT RELEVANT: Document doesn't help answer the query despite
  surface similarity

For this task, return documents that are RELEVANT or HIGHLY
RELEVANT only.

COMMON PITFALLS TO AVOID:
1. TOPICAL SIMILARITY != RELEVANCE
   Two documents about "machine learning" may address completely
   different aspects. The query might be about "ML for medical
   imaging" but doc is about "ML for NLP"
2. KEYWORD MATCHING != RELEVANCE
   Matching keywords doesn't mean the document answers the query.
   Context and the specific information need matter.
3. DOCUMENT QUALITY != RELEVANCE
   A poorly written document can still be relevant. A high-quality
   document on a different topic is not relevant.

WHEN IN DOUBT:
- Re-read the query and identify the core information need
- Ask: "If I were the user, would this document satisfy my search?"
- Lean toward precision: it's better to miss a marginally relevant
  doc than include an irrelevant one

---

USER:
Judge which documents are RELEVANT to the given search query.

A document is RELEVANT if it:
- Contains information that directly helps answer the query
- Would satisfy a user who submitted this search
- Provides substantive content on the query topic

A document is NOT RELEVANT if it:
- Only shares keywords but addresses a different information need
- Is about a related but distinct topic
- Mentions the query concepts tangentially without substantive
  coverage

Search Query: "{query_text}"

Candidate Documents:
{documents_text}

For each document, determine if it would SATISFY a user searching
for this query. Consider:
1. Does this document contain information the user is looking for?
2. Would returning this document be helpful for this specific query?
3. Is the relevance substantive or merely superficial keyword overlap?

Respond with ONLY the document numbers that are RELEVANT,
```

```
comma-separated (e.g., "0, 2, 5"). If none are relevant,
respond with "NONE".
```

## D. Examples and Coverage Analysis

### D.1. Example Semantic Cluster

Table 9 shows an example cluster capturing the semantic region of "fruit juices and their health effects," with entities spanning beverage types, specific juice varieties, and associated medical conditions.

### D.2. Cluster Visualization

Figure 6 visualizes the semantic clusters across the NFCorpus dataset.

**Semantic Communities: NFCorpus**

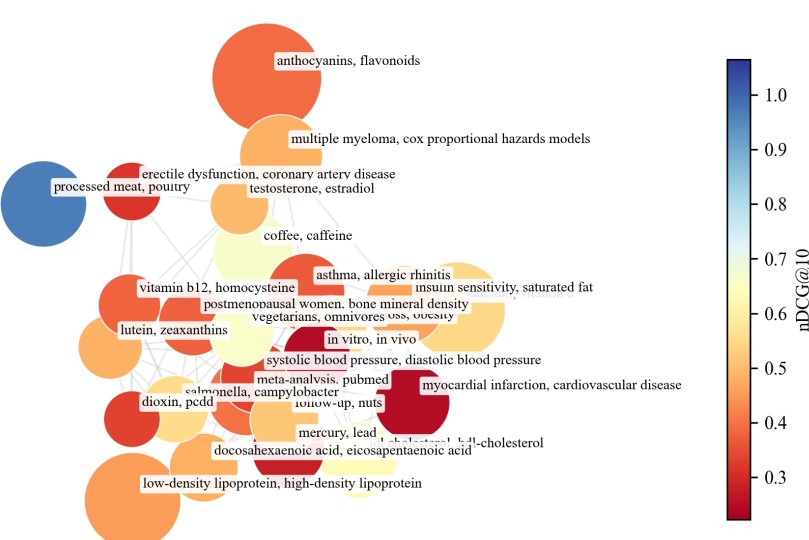

*Figure 6.* NFCorpus semantic cluster graph. Nodes represent entity clusters, node size indicates document count, and edges represent semantic similarity between clusters. Colors indicate avg in-cluster nDCG score.

### D.3. Example Generated Queries

Sample generated NFCorpus queries with strata values and relevant documents. Example queries show increase in Zero-Query Cluster coverage.

```
[Q1] "renal function differences vegans non-vegans"
    Docs (7): MED-5340, MED-3230, MED-3236, MED-5338, MED-934, MED-1458, MED-1456
    Delta: 0.530 | J: 0.159 | Clusters: 7 (6 new)

[Q2] "How do different vegetarian diets affect cholesterol and triglyceride levels compared to omnivorous diets in
     adults?"
    Docs (18): MED-1430, MED-4632, MED-4831, MED-5329, MED-1551, MED-2945, MED-4088, MED-4515, MED-1253, MED-1250,
        MED-5118, MED-4353, MED-4509, MED-5113, MED-2008, MED-1249, MED-4510, MED-1258
    Delta: 0.384 | J: 0.227 | Clusters: 15 (7 new)

[Q3] "glycemic response of different date varieties in healthy versus type 2 diabetic adults"
    Docs (2): MED-3897, MED-3907
    Delta: 0.429 | J: 0.444 | Clusters: 4 (4 new)
```

| Entity | Description |
| --- | --- |
| apple juice | A beverage made from the pressing and processing of apples |
| cranberry juice | A beverage made from cranberries, often studied for health benefits |
| fruit juice | Beverage made by extracting liquid from fruits |
| cranberries | A fruit known for its health benefits, particularly for urinary health |
| grapefruit | A citrus fruit known for its potential interactions with medications |
| pomegranate juice | Juice extracted from pomegranate, known for its health benefits |
| red wine | A type of wine made from dark-colored grape varieties |
| white wine | A type of wine made from green grapes |
| noni juice | Juice derived from noni fruit, marketed for health benefits |
| mangosteen juice | Juice extracted from mangosteen, associated with health claims |
| hibiscus tea | A herbal tea made from dried petals of hibiscus flowers |
| cherry juice | A beverage made from cherries, noted for health benefits |
| smoothie | A blended beverage typically made from fruits and vegetables |
| bladder infection | An infection of the urinary bladder (frequently studied in relation to cranberry juice) |
| *... and 53 more entities* | |

*Table 9.* Example Semantic Cluster: Fruit Juices & Beverages (Cluster 118, NFCorpus)

Sample generated FIQA queries with strata values and relevant documents. Example queries show increase in Zero-Query Cluster coverage.

```
[Q1] "reverse stock split investor impact"
    Docs (16): 579056, 507021, 229707, 477951, 388065, 304085, 502970, 513620, 237390, 581848, 159439, 129481,
        524940, 512914, 245654, 352130
    Delta: 0.587 | J: 0.031 | Clusters: 2 (2 new)

[Q2] "Chinese public opinion of Donald Trump during 2016-2020 presidential elections"
    Docs (4): 541283, 279360, 291836, 106268
    Delta: 0.800 | J: 0.333 | Clusters: 4 (3 new)

[Q3] "economic confirmation bias in academic research debates"
    Docs (8): 303978, 498180, 187090, 457235, 394096, 382709, 118092, 394357
    Delta: 0.707 | J: 0.069 | Clusters: 2 (1 new)
```

Sample generated SciDocs queries with strata values and relevant documents. Example queries show increase in Zero-Query Cluster coverage.

```
[Q1] "neural encoding auditory stimuli"
    Docs (10): 620b7d0d5e2ceeba58d808dc3d7b09a9fb57831c, 4934ac57123efbfbde139ec846452acd95520b40, 3
        c203e4687fab9bacc2adcff2cbcab2b6d7937dd, 6d3f38ea64c84d5ca1569fd73497e34525baa215, 8
        bfedec4b58a5339c7bd96e367a2ed3662996c84, 6042669265c201041c7523a21c00a2ab3360f6d1,
        ca579dbe53b33424abb13b35c5b7597ef19600ad, f18821d47b2bee62a7b3047ed7b71a5895214619,
        bdfb20d112069c106a9535407d107a43ebe6517f, 8fc062ac5cded5cbacb164e2b128f9914eb04727
    Delta: 0.327 | J: 0.361 | Clusters: 13 (1 new)

[Q2] "information technology strategic impact"
```

```
     Docs (16): 14316b885f65d2197ce8c6d4ab3ee61fdab052b8, d1909ee7543ab92a8d8aad81af6c589278654b48, 42540
          e75173dea14c343a567f9fc722db8609e7e, 65e58981f966a6d1b62c1f7d889ae3e1c0a864a1, 5
          f09cb313b6fb14877c6b5be79294faf1f4f7f02, 4c341889abc15b296c85dd5b4e1356c664f633f4, 7
          bad7050e0f447ea68c1ff5838cd5392726ca388, c13ee933b7ddccb4a1ccc7820c09fda8f32d1fb5, 588
          b9fae704c1964d639a5f87c3793a1ad354e69, d60fcb9b0ff0484151975c86c70c0cb314468ffb, 37
          c998ede7ec9eeef7016a206308081bce0fc414, 8326f86876adc98c72031de6c3d3d3fac0403175,
          eec44862b2d58434ca7706224bc0e9437a2bc791, ac8877b0e87625e26f52ab75e84c534a576b1e77, 6
          e07fcf8327a3f53f90f86ea86ca084d6733fb88, 176d486ccd79bdced91d1d42dbd3311a95e449de
     Delta: 0.386 | J: 0.078 | Clusters: 4 (1 new)

[Q3] "What are the TG13 guideline recommendations for biliary drainage timing and antimicrobial use in Grade II and
     III acute cholangitis patients?"
     Docs (1): e65881a89633b8d4955e9314e84b943e155da6a9
     Delta: 0.250 | J: 1.000 | Clusters: 3 (1 new)
```

## D.4. Zero-Query Cluster Analysis

Table 10 provides detailed examples of zero-query clusters identified in NFCorpus. These clusters represent methodological topics (statistical methods, clinical trial designs, and survey techniques) that collectively span over 1,700 documents but receive no evaluation queries in the original benchmark. This systematic gap means retrieval performance on methodological queries (e.g., "What is a Cox proportional hazards model?") remains entirely untested.

| Cluster Description | Ent. | Docs |
|---|---|---|
| Statistical methods (Cox regression, hazard ratios) | 104 | 470 |
| Multivariate analysis methods | 135 | 325 |
| Clinical trial methods (RCTs, double-blind) | 141 | 450 |
| Dietary assessment (food frequency questionnaire) | 81 | 192 |
| Survey methodologies | 70 | 89 |
| Healthy human participants | 79 | 155 |

*Table 10.* Zero-query clusters in NFCorpus span 1,700+ documents but have no evaluation queries.

# E. Additional Experiment Results

## E.1. Results on Additional Datasets

Figures 7-12 present performance distributions and system comparisons for FiQA and SciDocs datasets.

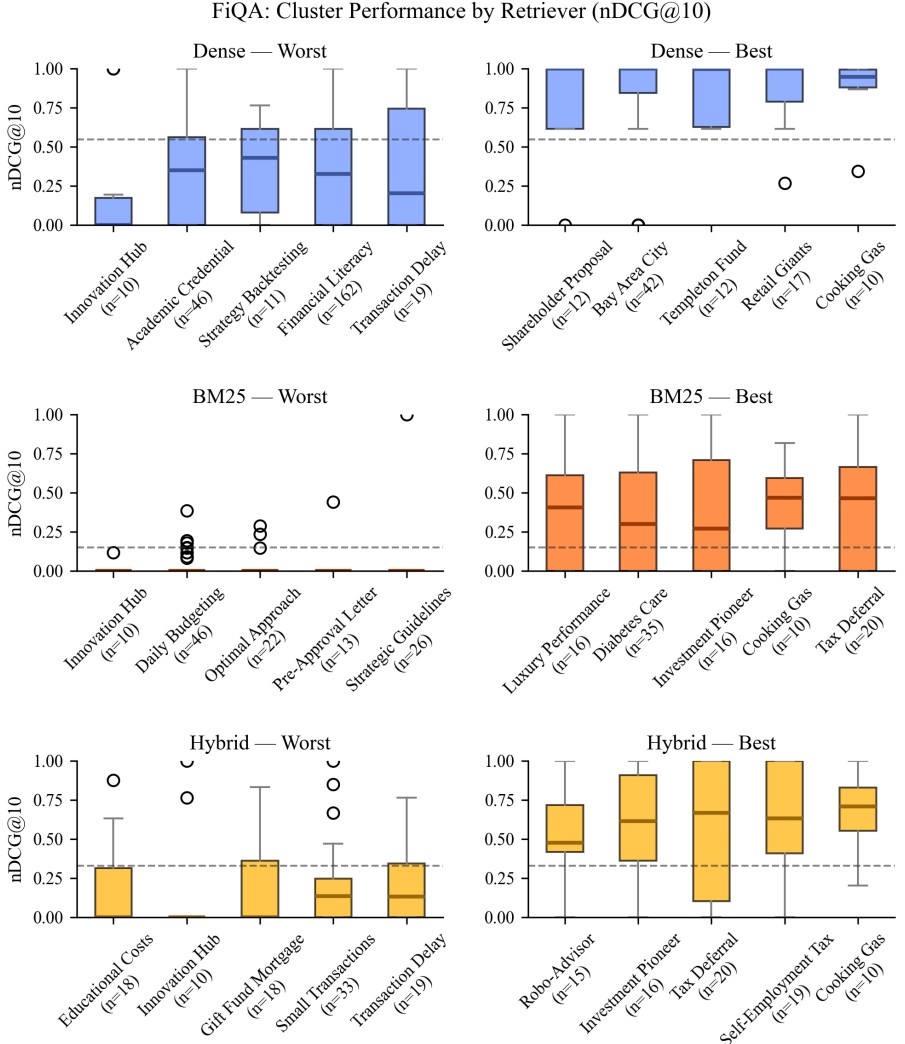

*Figure 7.* FiQA: Distribution of nDCG@10 scores across semantic clusters, showing per-cluster performance variance.

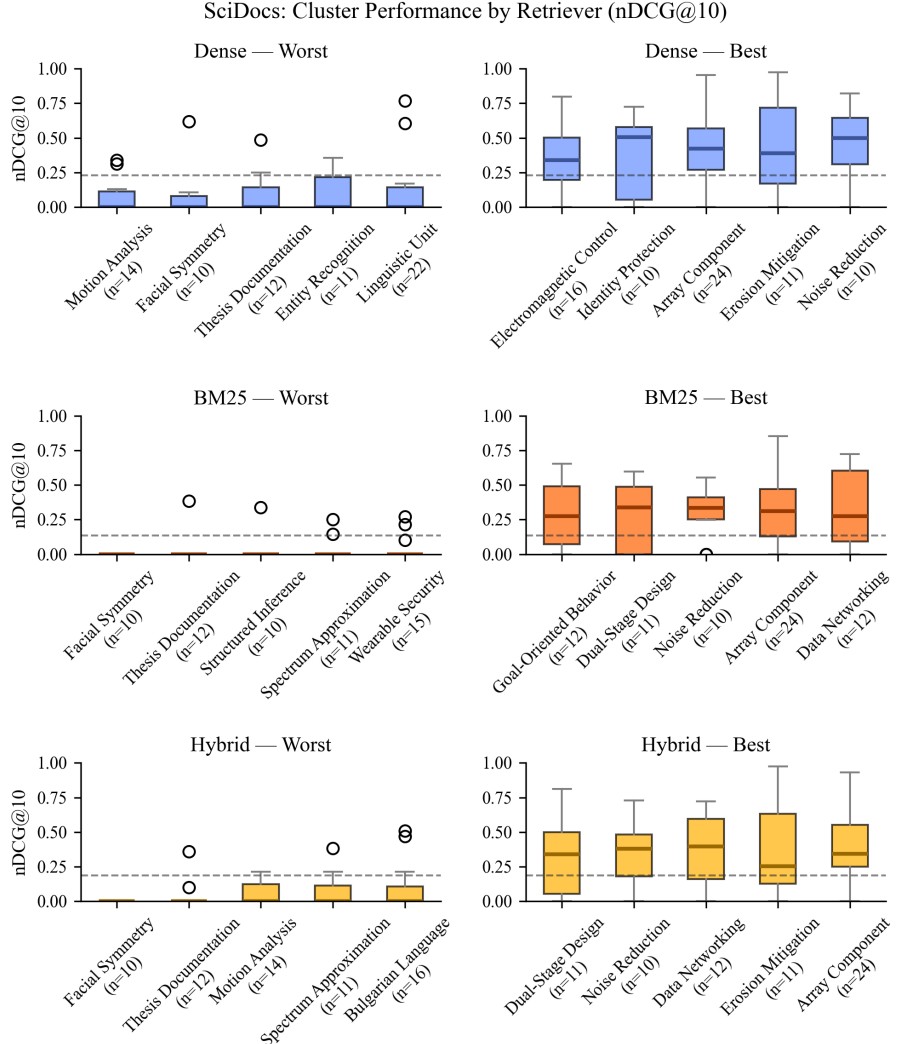

*Figure 8.* SciDocs: Distribution of nDCG@10 scores across semantic clusters, showing per-cluster performance variance.

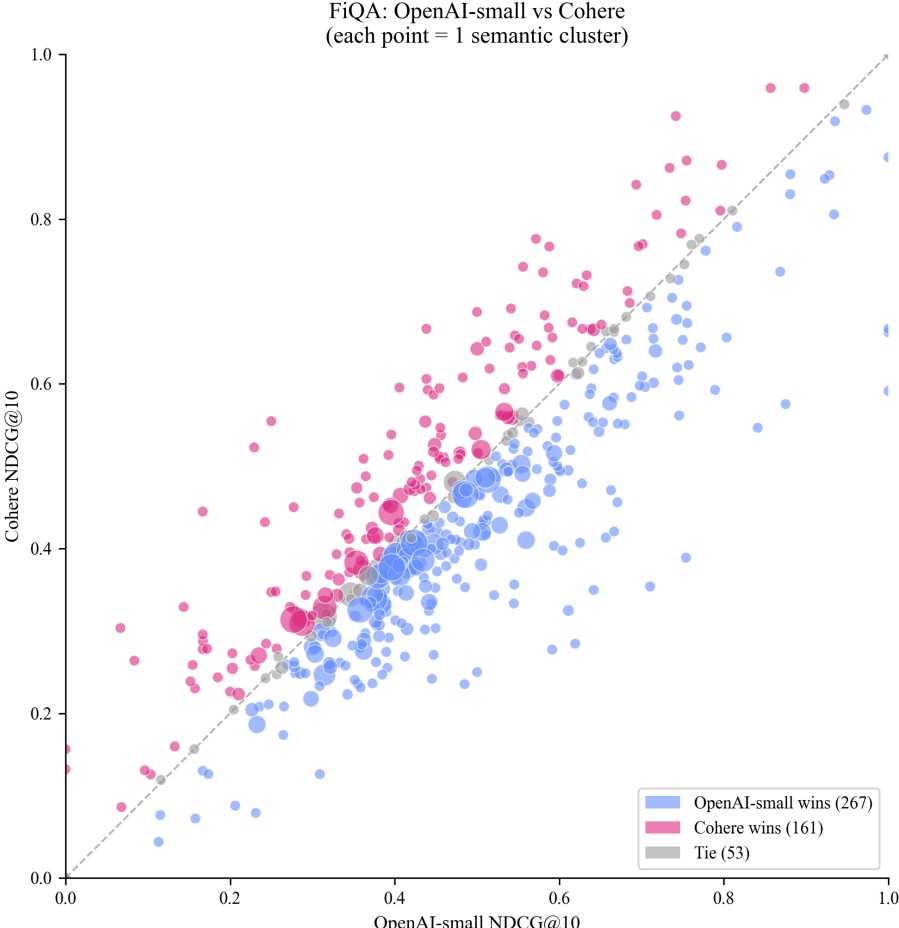

*Figure 9.* FiQA: Per-query performance comparison between retrieval systems, colored by semantic cluster.

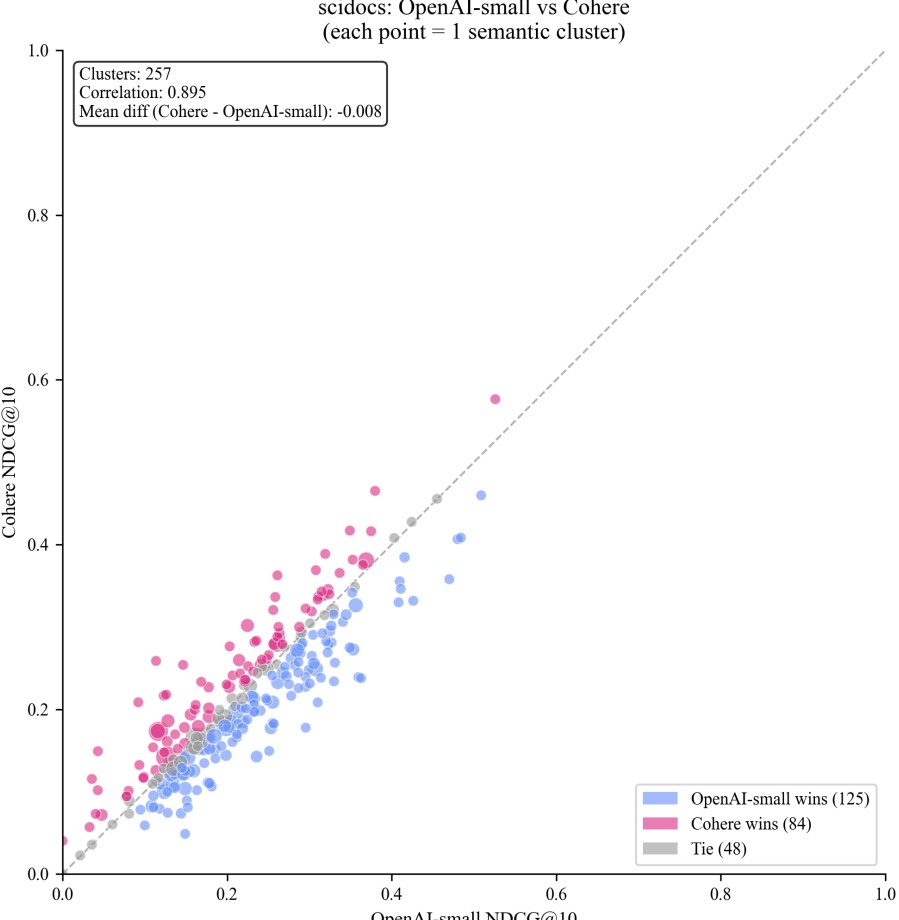

*Figure 10.* SciDocs: Per-query performance comparison between retrieval systems, colored by semantic cluster.

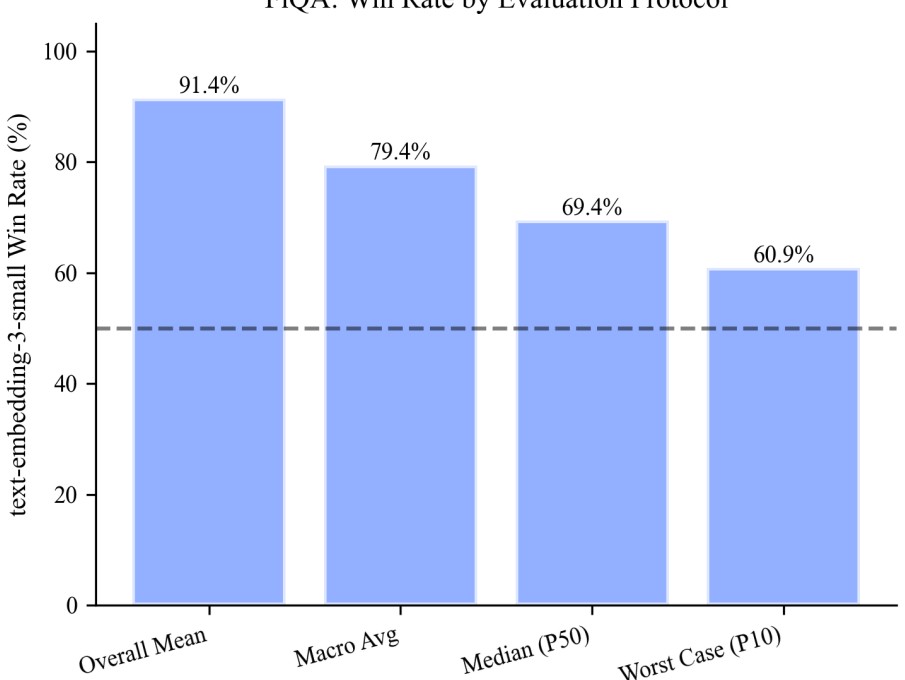

*Figure 11.* FiQA: Win rate comparison between retrieval systems across bootstrap samples (n=300).

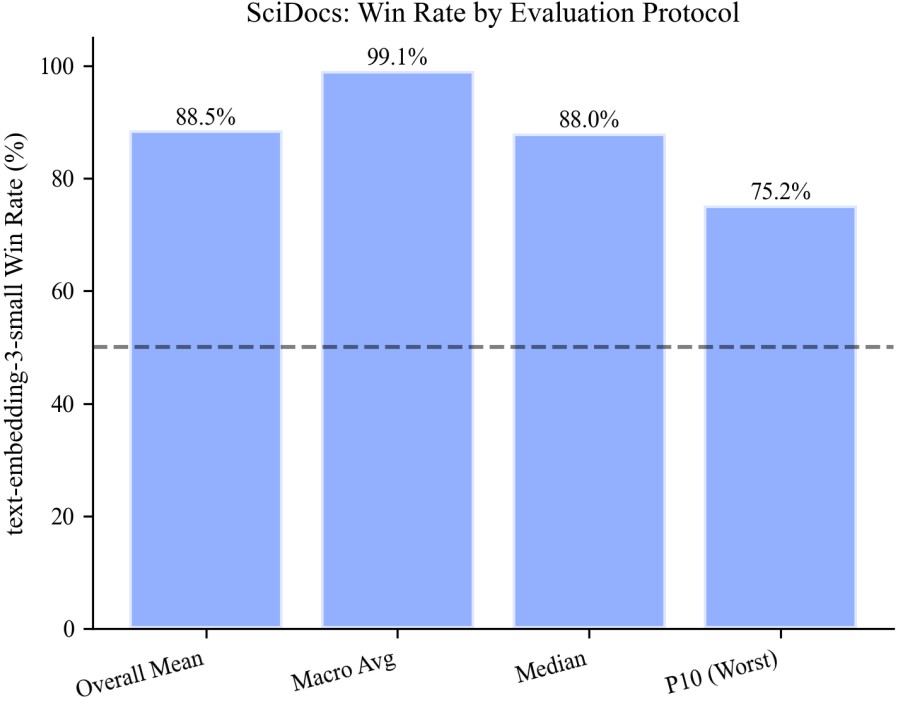

*Figure 12.* SciDocs: Win rate comparison between retrieval systems across bootstrap samples (n=300).

## E.2. Additional Interpretable Diagnostics

Beyond the corpus-level failure modes identified in the main paper (semantic-lexical disconnect, register mismatch), stratified evaluation reveals additional retriever-dependent failure patterns across datasets.

**Named Entity Mismatch (NFCorpus).** The *Pharmaceuticals and drug development* cluster (39 entities) exhibits a 75% zero-score rate across 8 queries. Sample entities include company names like "GlaxoSmithKline" and technical terms like "pharmaceuticalization." Queries in this cluster ask about drug development processes, but the named pharmaceutical companies mentioned in corpus documents rarely appear in natural user queries. This represents a retriever failure where proper noun specificity in documents does not match the more generic query formulations.

**High-Frequency Practical Topics (FiQA).** In FiQA, clusters representing common personal finance topics achieve disproportionately poor retrieval performance:

- **Retirement investment strategies** (62 entities, nDCG = 0.23): 66.7% zero-score rate despite targeting common concepts like "target date retirement funds."

- **Language and terminology in finance** (144 entities, nDCG = 0.17): 62.5% zero-score rate for queries about contextual meaning and financial terminology.

- **Customization and personalization** (nDCG = 0.11): Abstract concepts like "personalized gifts" and "features" that are too generic to embed discriminatively.

These high-frequency, practical questions fail because common financial terminology may be too generic in the embedding space to discriminate between relevant and irrelevant documents.

**Technical Algorithm Names (SciDocs).** In SciDocs, clusters containing highly specialized algorithm names achieve near-zero performance:

- **Optimization algorithms** (424 entities, nDCG = 0.07): 75% zero-score rate with entities like "NelderMeadAlgorithm" and "alternating direction method of multipliers."

- **Sentiment analysis** (308 entities, nDCG = 0.07): 77.8% zero-score rate despite covering a well-established NLP subfield.

- **Digital healthcare innovations** (268 entities, nDCG = 0.02): 80% zero-score rate for "quantified self" and "healthcare innovation" topics.

The failure despite exact terminology matches suggests that these technical terms may be underrepresented in the embedding model's training data, or that the citation-based relevance judgments in SciDocs create a mismatch between lexical similarity and annotated relevance.

**High-Performing Clusters.** For contrast, we identify consistently high-performing clusters:

- **NFCorpus**: Viruses linked to obesity (nDCG = 0.81), MRSA strains (nDCG = 0.77), gluten-related disorders (nDCG = 0.76)

- **FiQA**: Tax incentives (nDCG = 0.97), corporate scandals (nDCG = 0.96), aviation industry (nDCG = 0.96)

- **SciDocs**: Algebraic structures in programming (nDCG = 0.72), amplifier design (nDCG = 0.72), wearable health monitoring (nDCG = 0.57)

These clusters succeed because queries and documents use consistent domain terminology with distinctive named entities, validating that retrieval works well when lexical-semantic alignment holds.

# F. Additional Related Work

The evaluation of Retrieval-Augmented Generation (RAG) has evolved from lexical n-gram matching to principled methodologies that isolate the performance of retrieval and generation components.

**IR Evaluation.** Modern RAG evaluation inherits methodology from information retrieval research such as the Text REtrieval Conference (TREC) which established standardized test collections, relevance judgments, and evaluation metrics (Voorhees & Harman, 2005). These efforts culminated in unified Question Answering (QA) benchmarks like BEIR (Thakur et al., 2021), which aggregates 18 diverse and foundational sets like NQ, HotpotQA, TriviaQA, and MS MARCO (Kwiatkowski et al., 2019; Joshi et al., 2017; Nguyen et al., 2016; Yang et al., 2018).

**RAG Benchmarks.** Modern RAG-specific benchmarks extend these foundations to target specific RAG failure modes. MultiHop-RAG requires synthesis across multiple documents (Tang & Yang, 2024). CRAG provides question-answer pairs spanning five domains with varying entity popularity, temporal dynamics, and complexity (Yang et al., 2024). For corpus-level reasoning, GlobalQA introduces aggregation tasks such as counting, sorting, and top-k extraction over entire collections (Luo et al., 2025). Long$^2$RAG addresses long-context scenarios with the Key Point Recall metric for evaluating generation (Qi et al., 2024), MT-RAG benchmarks multi-turn conversational RAG with human-generated dialogues (IBM Research, 2025), mmRAG (Xu et al., 2025) tackles heterogeneous knowledge sources, and QE-RAG (Zhang et al., 2025) evaluates robustness to query entry errors such as typos and keyboard proximity mistakes.

**Query Taxonomies and Curation.** Principled curation by defining taxonomies that reflect real-world difficulty. Teixeira de Lima et al. (2025) proposed a taxonomy classifying context-query pairs into predefined categories such as fact_single, summary, reasoning, and unanswerable categories to identify specific retrieval challenges. The CRUD-RAG benchmark further organizes tasks into Create, Read, Update, and Delete operations to evaluate RAG systems in scenarios beyond standard question-answering, such as text continuation and hallucination modification (Lyu et al., 2025).

**Corpus Stratification.** Works like GraphRAG (Edge et al., 2025), RAPTOR (Sarthi et al., 2024), and HippoRAG (Gutierrez et al., 2024) have explored corpus stratification to improve retrieval efficiency and quality. GraphRAG's stratification is most similar to ours as it constructs clusters using the Leiden algorithm (Traag et al., 2019) after LLM-prompt based entity and relationship extraction. Our approach differs in that relationships are detected directly in the semantic space and the stratification is used to inform benchmark diagnosis and repair rather than retrieval optimization.

**Synthetic Data Generation (SDG).** To overcome the bottleneck of manual annotation, frameworks like RAGEval and Long$^2$RAG employ schema-based pipelines to generate grounded documents, questions, and answers (Zhu et al., 2025; Qi et al., 2024). Similarly, DataMorgana provides a customizable framework to control the lexical, syntactic, and semantic diversity of generated benchmarks, ensuring they reflect the expected traffic of enterprise applications (Filice et al., 2025).

**Efficient Evaluation under Budget Constraints.** Complementary work reduces annotation costs via strategic item selection: active learning for IR test collections (Rahman et al., 2020), minimal test collections (Carterette et al., 2006), and compact LLM benchmarks (Polo et al., 2024). These approaches select items based on model uncertainty or statistical informativeness; in contrast, our framework prioritizes corpus-informed coverage independent of the systems being evaluated.

