# OpenReview forum: "Coverage, Not Averages: Semantic Stratification for Trustworthy Retrieval Evaluation"
_ICML.cc/2026/Conference — ICML 2026 regular_

### Official Review · Reviewer_4xoX · 2026-03-10

**Soundness:** 3
**Presentation:** 3
**Significance:** 3
**Originality:** 2
**Overall Recommendation:** 4
**Confidence:** 4

**Summary:**

This paper addresses a fundamental problem in RAG evaluation: standard benchmarks rely on heuristically assembled query sets that systematically under-cover large regions of the document corpus's semantic space, and these under-covered regions tend to be where retrieval performance is weakest. This creates an upward bias in aggregate metrics that can distort system comparisons. The paper formalizes retrieval evaluation as a statistical estimation problem under structured heterogeneity, showing that naive empirical averaging over non-representative query sets yields biased performance estimates. It proposes semantic stratification: organizing the corpus into interpretable entity-based clusters via LLM-extracted entities, a semantic similarity graph, and the Leiden community detection algorithm, then defining retrieval regimes along semantic and structural dimensions. An iterative coverage-aware query generation procedure fills identified gaps. Experiments on three BEIR datasets validate that standard benchmarks leave 40–50% of semantic regions untested, and that stratified evaluation reveals failure modes invisible to aggregate metrics.

**Compliance With Llm Reviewing Policy:**

Affirmed.

**Final Justification:**

The paper addresses an important problem of existing benchmarks and proposes a solution with a well-motivated stratification design. The rebuttal addressed my main concerns. I raised my score to 4 since I think the overall impact is modest.

**Key Questions For Authors:**

Q1. The system comparison in Section 5.5 involves two models with very similar overall performance, making rank reversal under different aggregation protocols unsurprising. Can the authors provide an example — either from their experiments or a constructed case — where stratified evaluation corrects a ranking between systems with a clearer overall performance difference?

Q2. The resolution parameter γ is tuned per-dataset (25, 40, and 10 for NFCorpus, FiQA, and SciDocs respectively) without a principled automated selection criterion. How sensitive are the downstream coverage statistics and evaluation conclusions to this choice? What guidance would you offer a practitioner applying this framework to a new corpus?

Q3. Since relevance judgments are produced entirely by LLM filtering, how reliable are these compared to human annotations? Have the authors validated LLM relevance judgments against human labels on any subset of the generated queries?

Q4. The pipeline involves LLM calls for entity extraction across all documents, graph construction, Leiden clustering, and iterative LLM-based query generation and relevance filtering. What is the approximate computational cost for the datasets used, and how does this scale to larger corpora (e.g., MS MARCO with millions of documents)?

**Limitations:**

yes

**Strengths And Weaknesses:**

**Strengths**

**S1. Important and underexplored problem.** This study claims to present a critical issue that receives far less attention than retrieval architecture or generation quality. The statistical formalization demonstrates that coverage gaps produce irreducible bias regardless of evaluation set size, which is a strong result. The empirical finding that under-covered regions are systematically those with the lowest retrieval performance makes the practical stakes concrete.

**S2. Well-motivated stratification design.** Overall, the authors outline a central context in which the two structural signals — relevance dispersion (Δ) and query-document alignment (J) — are jointly justified. Table 3 validates empirically that these signals partition queries into genuinely distinct difficulty regimes. The finding that dispersion primarily governs ranking quality (nDCG) while alignment primarily governs coverage (Recall) is interpretable and actionable.

**S3. Meaningful coverage improvements demonstrated.** Table 4 shows substantial gains: NFCorpus minimum semantic coverage increases from 51% to 90%, with zero-query clusters dropping from 182 to 36. The identification of specific zero-query cluster types spanning 1,700+ documents (Table 8) compellingly demonstrates that coverage gaps are systematic rather than incidental.


**Weaknesses**

**W1. Practical impact on method ranking is insufficiently demonstrated.** The most important practical question is not whether aggregate metrics are technically biased, but whether this bias actually changes which methods are selected. The paper addresses this only partially through a comparison of two similarly performing systems (OpenAI vs. Cohere), where rank reversal under different aggregation protocols is unsurprising. The paper does not demonstrate that stratified evaluation would correct a ranking between a genuinely better and a genuinely worse system that aggregate metrics incorrectly ordered. Without such a demonstration, it remains unclear whether the identified coverage gap constitutes a practical problem or primarily a theoretical concern.

**W2. Insufficient engagement with related literature.** The problem of benchmark quality and query selection bias has a long history in both and NLP that the paper does not adequately engage with. The paper also does not connect to slice-based evaluation from the fairness and robustness literature, where subgroup-level performance analysis is standard. More thorough positioning would clarify what is genuinely novel versus what applies known statistical ideas to a new domain.

**W3. Overlap with prior work understated.** While the paper notes that GraphRAG, RAPTOR, and HippoRAG use corpus stratification for retrieval optimization rather than evaluation, the methodological overlap is significant. GraphRAG uses the same Leiden clustering on LLM-extracted entities — the primary difference is downstream use. The paper should more explicitly characterize which pipeline components are novel contributions versus applications of existing techniques to a new problem setting.

**W4. LLM dependence introduces unquantified biases.** The framework relies on LLMs for entity extraction, query generation, entity assignment, and relevance filtering. LLM-generated queries may over-represent topics salient in the model's training data, partially reproducing the coverage biases the paper aims to fix. LLM-based relevance judgments are known to exhibit systematic biases (length, positional, fluency preferences). The paper acknowledges these issues in Section 7 but does not quantify them or validate against human annotations, making it difficult to assess whether the stratified sets are genuinely more reliable.

**W5. Limited novelty for an ICML submission.** The statistical formalization applies standard stratified sampling theory without substantial theoretical development beyond the bias decomposition. The semantic structure construction is a pipeline of existing components (LLM extraction → embedding → nearest-neighbor graph → Leiden clustering). The coverage-aware generation relies primarily on prompt engineering rather than novel algorithms. The work may be better suited for a benchmarking or evaluation-focused venue, where dataset and evaluation contributions are the primary focus.

---

> ### Author Rebuttal · Authors · 2026-03-31
>
> We thank the reviewer for their thorough and constructive review, and we appreciate the opportunity to solidify our positioning.
>
> **[W1 & Q1] Practical impact on method ranking**
>
> We acknowledge that a more dramatic ranking correction between clearly separated systems would strengthen this analysis. However, we note that the case of similarly performing systems is precisely where evaluation protocol matters most as this is when practitioners are most likely to make the wrong choice, and when the evaluation methodology is most load-bearing. The Figure 5 result shows that *text-embedding-3-small*'s win rate against Cohere's *embed-english-v3.0* shifts from 32.4% (overall mean) to 85.8% (worst-case P10). A practitioner who cares about tail performance would make the opposite decision from one optimizing for the average case.
>
> More broadly, when performance varies 3x across semantic regions, a single ranking obscures more than it reveals. System A may be better for medical terminology and worse for statistical methods. Stratified evaluation makes this visible, enabling practitioners to choose based on what matters for their use case rather than relying on an aggregate that may not reflect their priorities.
>
> **[W2, W3 & W5] Novelty, related work, and overlap with prior methods**
>
> The core contribution of this paper is the perspective that retrieval evaluation should be *stratified* with respect to the corpus semantic structure, and the formalization of why this matters (Sec. 3). The bias and variance decomposition (Eqs. 1-5) provides a theoretical foundation for why coverage-aware evaluation is necessary, not merely desirable, and the empirical finding that 40-50% of semantic regions go untested in standard benchmarks is new to the literature.
>
> We have expanded the related work to connect with slice-based evaluation from the fairness/robustness literature. Our distinction is that slice-based approaches define subgroups based on query-level attributes, whereas our strata are grounded in corpus structure and are method-independent.
>
> On GraphRAG: while both methods use LLM-extracted entities and Leiden clustering, the overlap is at the component level. GraphRAG builds hierarchical communities to optimize retrieval, while we build a flat semantic partition to diagnose evaluation. This is a fundamentally different goal that requires method-independent strata (Sec. 3), a constraint absent from GraphRAG. The graph construction also differs: GraphRAG uses LLM calls to determine entity relationships, which is expensive and model-dependent, whereas we embed entities via a dense model and construct a KNN similarity graph. To be explicit about novelty: entity extraction and Leiden clustering are existing techniques applied to a new problem; the structural signals ($\Delta$, J), coverage metrics (MSC, SCC, ZQC), stratified estimation framework, and coverage-aware curation algorithm (Algorithm 1) are novel contributions with no analogue in GraphRAG.
>
> We believe evaluation methodology is within ICML's scope. The venue has a strong tradition of papers on benchmarking, measurement, and evaluation (e.g., Polo et al., 2024, published at ICML). We will revise to make the positioning clearer.
>
> **[W4 & Q2] LLM dependence and resolution sensitivity**
>
> The extracted entities are predominantly well-known domain concepts and named entities where modern LLMs show strong agreement, and our normalization pipeline further reduces surface-level variation. Our resolution ablation (table in Reviewer snb4 response) demonstrates robustness across $\gamma \in$ {10, 15, 25, 40, 60}, producing 155 to 669 clusters with all diagnostic conclusions holding. Selecting gamma requires only inspecting cluster coherence at the desired granularity. Re-clustering on cached embeddings is a token-free process, making it practical to try multiple values. The full resolution ablation and practitioner guidance will be included in the camera-ready.
>
>
> **[Q3] Reliability of LLM relevance judgments**
>
> We note indirect evidence of judgment quality in our zero-query cluster probe (see Reviewer Ensk Q1 for full results). We observed that the LLM-generated qrels produced retrieval scores consistent with expectations: high nDCG on topically specific clusters (e.g., cancer cell lines, 0.87 hybrid) and lower nDCG on diffuse methodological clusters (e.g., statistical methods, 0.48 hybrid). If the relevance judgments were unreliable, we would not expect this coherent pattern. We agree that direct validation against human annotations is important and discuss this in Sec. 7.
>
> **[Q4] Approximate computational cost**
>
> See our resolution ablation above for the full argument (Reviewer snb4 Q4). In short, the complete upper bound costs for the full pipeline state in our paper is under $1000, even for large datasets comprising over 8 million documents, such as MS MARCO. We believe this is a reasonable cost for running our analysis in a production setting.

---

> > ### Author Rebuttal · Reviewer_4xoX · 2026-04-02
> >
> > Thanks for your clarifications. I raised my score.

---

### Official Review · Reviewer_Ensk · 2026-03-11

**Soundness:** 3
**Presentation:** 2
**Significance:** 2
**Originality:** 3
**Overall Recommendation:** 4
**Confidence:** 3

**Summary:**

This paper argues that existing retrieval evaluation suffers from biased results because the semantic distribution of queries is incomplete and imbalanced, as queries exhibit structural heterogeneity across retrieval regimes. To address this issue, the authors propose a Stratified Evaluation framework that organizes the corpus into a semantic space by clustering documents based on entities. Queries are then systematically constructed to ensure broader semantic coverage across this space. Experiments on three datasets evaluate three retrieval methods using the proposed framework, revealing several common patterns in retrieval performance.

**Compliance With Llm Reviewing Policy:**

Affirmed.

**Final Justification:**

The rebuttal has addressed most of my concerns, which has led me to increase my score to 4.

**Key Questions For Authors:**

1. Which queries mainly contribute to the observed failures?
Are the poorly performing queries mainly from the original dataset or from the constructed queries introduced by stratified evaluation? In particular, how do the queries that fill the Zero-Query Clusters perform compared with the original queries?
2. Are the constructed queries distributionally different from the original ones?
Do the generated queries differ significantly from the original queries in terms of style, length, or wording? If so, such differences may also affect retrieval performance.
3. How important is semantic coverage given the generally low retrieval performance on some benchmarks?
For example, on NFCorpus, retrieval performance is already relatively low (with the best NDCG@10 on the BEIR leaderboard around 55.74). In this case, how much does incomplete semantic coverage affect the reliability of model evaluation?
4. Would the conclusions hold with more models and larger datasets?
The current evaluation only includes two retrieval methods and relatively small datasets. It would be helpful to examine whether the same observations hold when evaluating more diverse retrieval models and larger benchmarks.

**Limitations:**

yes

**Strengths And Weaknesses:**

Strengths
1. Meaningful perspective. The paper highlights query coverage as an important issue in retrieval evaluation and provides a clear analysis of how existing benchmarks suffer from imbalanced semantic distributions.
2. Clear empirical evidence. The experiments provide clear empirical evidence of semantic coverage gaps in standard benchmarks and their impact on aggregate evaluation metrics.

Weaknesses
1. Unclear experimental conclusions. The experimental section is somewhat fragmented, and the key findings are not clearly summarized.
2. Limited evaluation methods. Only two retrieval methods (BM25 and text-embedding-3-large) are evaluated, which limits the generality of the conclusions.
3. Limited scale of evaluation datasets. The datasets used contain relatively small document collections and few queries. It is unclear whether similar observations would hold on larger benchmarks such as MS MARCO.

---

> ### Author Rebuttal · Authors · 2026-03-31
>
> We thank the reviewer for their timely feedback. We have conducted a diagnostic probe of zero-query clusters to directly address the concerns raised. Please see our responses below.
>
> **[W1] Unclear experimental conclusions**
>
> We have revised Sec. 5 to more clearly summarize the key findings at the end of each subsection. The core results are: (1) semantic strata capture meaningful structure with up to 3x performance variation across clusters (Sec. 5.2), (2) existing benchmarks leave 40-50% of semantic regions untested (Sec. 5.3), (3) stratified analysis reveals failure modes invisible to aggregate metrics (Sec. 5.4), and (4) stratified evaluation directly affects real system selection decisions (Sec. 5.5).
>
> **[W2] Limited evaluation methods**
>
> We would like to clarify that we evaluate three retrieval paradigms, not two (Sec. 5.1): dense retrieval, BM25, and hybrid via Reciprocal Rank Fusion. These operate on entirely different principles (lexical vs. learned semantics). The consistent stratum-level patterns across all three (Figure 3) indicate that our semantic structure captures genuine corpus properties.
>
> **[W3] Limited scale of evaluation datasets**
>
> We selected specialized domain datasets (medical, finance, scientific) where semantic coverage is particularly important. The consistent patterns across all three datasets suggest generalizability. We have also run the pipeline on HotpotQA (5.2M documents). The framework successfully produces coherent clusters, though at this scale clusters tend to be very large, reducing diagnostic granularity. The framework is best suited for domain-specific corpora where semantic regions are well-differentiated, which is also where coverage gaps matter most in practice (e.g., enterprise RAG over proprietary corpora). General-purpose corpora like full Wikipedia present different challenges that we discuss in Sec. 7.
>
> **[Q1] Which queries contribute to observed failures? How do zero-query clusters perform?**
>
> The framework reveals two categories of failures, both analyzed on the original BEIR queries. First, **retriever-specific failures**: Figure 3 shows that each retrieval paradigm has distinct worst-performing clusters. Conversely, *Nutrition Expert* is the lowest-performing stratum for all three retrievers, indicating systematic retrieval difficulty. Second, **corpus-level failures**: Appendix E identifies failure modes where retrieval success is impossible due to corpus construction issues.
>
> Regarding zero-query clusters: to directly test what happens in these untested regions, we generated queries targeting two zero-query clusters in NFCorpus and evaluated all three retrieval methods (nDCG@10):
>
> | Zero-Query Cluster | Dense | BM25 | Hybrid |
> |--------------------|-------|------|--------|
> | Cancer cell lines | 0.795 | 0.854 | 0.872 |
> | Statistical methods | 0.365 | 0.336 | 0.483 |
>
> These results illustrate precisely why coverage matters. *Cancer cell lines* achieves strong retrieval across all methods, yet this performance is entirely absent from the benchmark's aggregate score. Also, *statistical methods* is a genuine blind spot where all retrievers struggle. The two clusters also reveal different retriever dynamics: BM25 outperforms dense on cancer cell lines, while hybrid provides the largest boost on statistical methods.
>
> **[Q2] Are constructed queries distributionally different?**
>
> Yes, the generated queries target semantic regions that the original queries don’t reach, so distributional differences are expected. Within each stratum, we encourage diversity in query length and specificity to better match real-world search patterns (details in Appendix C). The goal is not to replicate the original query distribution but to exercise untested semantic regions. We agree that a more thorough analysis of how query style interacts with retrieval performance across strata is a valuable direction for future work.
>
> **[Q3] Importance of coverage in benchmarks with low baseline performance?**
>
> The stratified lens transforms a low aggregate score from a dead end into an actionable diagnostic. On NFCorpus, Figure 3 reveals that retrieval works remarkably well in several semantic regions (nDCG@10 > 0.8 for clusters like gluten-related disorders and MRSA strains), while failing in others. A practitioner seeing only the aggregate nDCG@10 of ~0.34 might conclude the retriever is uniformly mediocre, but stratified evaluation shows instead where it succeeds and where it needs improvement, motivating targeted interventions rather than wholesale system changes.
>
> **[Q4] Would conclusions hold with more models and larger datasets?**
>
> We evaluate three retrieval paradigms (dense, BM25, hybrid) and observe consistent patterns across all three. The structural findings (coverage gaps, within-stratum homogeneity, structural signal predictiveness) are properties of corpus-query relationships, not specific models. We agree that larger-scale validation is valuable and discuss this in Sec. 7.

---

> > ### Author Rebuttal · Reviewer_Ensk · 2026-04-03
> >
> > Thanks for the clarifications. I have raised my score accordingly.

---

### Official Review · Reviewer_VJKp · 2026-03-11

**Soundness:** 2
**Presentation:** 3
**Significance:** 3
**Originality:** 2
**Overall Recommendation:** 4
**Confidence:** 3

**Summary:**

This paper investigates a reliability problem in retrieval and RAG evaluation. Benchmark query sets may provide poor semantic coverage of the underlying corpus, causing aggregate retrieval metrics to misrepresent system behavior. The authors formalize evaluation as a stratified estimation problem, arguing that when query space is heterogeneous across retrieval regimes, naive aggregate estimates can be biased or overconfident. To address this, the authors propose a semantic stratification framework that constructs an interpretable corpus-level semantic space using LLM-extracted entities and clustering, defines semantic and structural strata, and fills uncovered regions with coverage-aware synthetic query generation. Experiments on several BEIR datasets and retrieval methods show substantial coverage gaps in standard benchmarks, strong variation in retrieval performance across strata, and improved diagnostic transparency under the proposed evaluation protocol.

**Compliance With Llm Reviewing Policy:**

Affirmed.

**Final Justification:**

Most of my concerns have been resolved. Although I still believe that some analysis of the end-to-end RAG results should be added, which may bring new insights, this does not affect the main contribution of the paper.

**Key Questions For Authors:**

1. What is the precise estimand of the proposed evaluation protocol when the true deployment query distribution is unknown? Please clarify under what assumptions the method should be interpreted as reducing bias rather than simply enforcing broader corpus coverage.

2. How reliable are the LLM-based entity extraction, query generation, entity assignment, and relevance verification stages? Can you provide validation against human judgments or at least robustness analyses across models?

3. Can you disentangle the effects of stratified evaluation from those of synthetic query augmentation? For example, can you compare same query pool with different aggregation/reporting schemes, or augmented queries without semantic stratification?

4. Since the paper is positioned in the context of RAG, can you clarify how this framework would extend to end-to-end RAG evaluation beyond retrieval alone? In particular, do better semantic coverage and stratum-level retrieval diagnostics translate into measurable differences in generation metrics such as faithfulness, answer relevance, or robustness?

**Limitations:**

yes

**Strengths And Weaknesses:**

S1. The paper addresses an underexplored problem in retrieval/RAG evaluation, namely the mismatch between benchmark query composition and corpus semantic coverage.

S2. The statistical framing of evaluation under structured heterogeneity is intuitive and helps motivate why aggregate metrics can hide systematic failure modes.

S3. The proposed framework is original in combining corpus-level semantic clustering, structural difficulty signals, and coverage-aware dataset curation into a coherent evaluation methodology.

S4. The experiments across multiple datasets and retrieval paradigms provide useful evidence that semantic coverage gaps are real and that stratum-level analysis can reveal practically meaningful diagnostics.

W1. The strongest claims about bias reduction and trustworthy estimation are not fully supported. Because the true deployment query distribution is unknown and the method effectively replaces it with a corpus-induced proxy.

W2. The pipeline depends heavily on LLM-based entity extraction, query generation, entity assignment, and relevance verification, but the paper provides limited validation or sensitivity analysis for these components.

W3. The empirical comparison conflates several factors: stratified reporting, synthetic query augmentation, and coverage-aware sampling. This makes it difficult to isolate which component is responsible for the observed gains.

W4. Although the paper mentions RAG, the evaluation is limited to the retrieval component only. It does not analyze how the proposed stratified framework affects end-to-end generation quality of RAG.

---

> ### Author Rebuttal · Authors · 2026-03-31
>
> We thank the reviewer for their detailed feedback which directly motivated the additional analysis below.
>
> **W1 & Q1: Unknown query distribution, $Q$, and its effects on estimation**
>
> We would like to restate that the estimand is not the population mean mu. As we argue in Sec. 3, $\mu = \sum_{k=1}^K w_k\mu_k$ depends on unknown deployment weights $w_k$ and is therefore ill-defined in offline evaluation settings. The bias analysis in Eq. (4) is included to explain *why* aggregate metrics are unreliable, not to claim that our method recovers a better estimate of $\mu$.
>
> Our framework targets the regime-level conditional means ${\mu_k} = E[\phi(q; \pi) | q \in S_k]$ as the evaluation output. These are well-defined quantities that characterize system behavior within coherent semantic regions, independent of any assumed query distribution. The evaluation output is a performance *profile*, not a scalar. Conditioning on a corpus-defined semantic structure is not an arbitrary proxy; the corpus defines the space of information a retrieval system can serve. A semantic region spanning 450 documents on clinical trial methods (Table 8) represents real retrieval scenarios regardless of how frequently users happen to query about them. If a system fails entirely on such a region, that is a meaningful finding that aggregate evaluation hides.
>
> The key assumption is that corpus-derived semantic regions correspond to meaningfully different retrieval regimes, which we validate empirically: strata exhibit internal homogeneity (Table 2), structural signals predict performance (Table 3), and performance varies up to 3x across strata (Figure 3).
>
> **W2 & Q2: Sensitivity and robustness analysis**
>
> See our resolution ablation above for the full argument (Reviewer snb4 Q3). In short, the framework's conclusions are robust to a 4x change in cluster granularity, suggesting robustness to moderate entity set perturbations. The extracted entities are well-known domain concepts and named entities where modern LLMs show strong agreement. We acknowledge full pipeline model validation as valuable future work.
>
> **W3 & Q3: Conflating factors in results section**
>
> This is a fair concern, and we want to clarify that these components are in fact evaluated separately in the paper. (1) Stratified reporting is evaluated entirely on original BEIR queries with no synthetic augmentation. Table 2 (within-stratum homogeneity), Table 3 (structural signal effects), Figure 3 (per-cluster performance variation), and Figure 5 (system comparison sensitivity) all use the existing benchmark query sets analyzed through our stratification framework. The finding that strata exhibit significant performance variation and that system rankings flip under different aggregation protocols requires no query generation at all. (2) Coverage diagnosis (Table 4, "Aggregate" column) identifies gaps in existing benchmarks (e.g., NFCorpus achieving only 51% MSC) using only the original queries mapped into our semantic structure. (3) Coverage-aware query generation (Table 4, "Stratified" column) is presented as a remediation tool applied after the diagnosis, and is  evaluated on coverage metrics (MSC, SCC, ZQC), not retrieval performance. We do not claim that generated queries improve retrieval, only that they improve evaluation coverage.
>
> We will revise the camera-ready to make this separation more explicit.
>
> **W4 & Q4: Extension to end-to-end RAG evaluation**
>
> We acknowledge that our experiments focus on the retrieval component. This is intentional. As motivated in Sec. 1, retrieval mediates all access to external knowledge in RAG. Our framework is motivated by enterprise settings where the corpus is proprietary and domain-specific. In this environment, retrieval quality is the primary bottleneck, as the generator has no prior knowledge of the corpus content and relies entirely on retrieved documents for grounding. The framework identifies semantic regions where retrieval systematically fails (Sec. 5.4), which represent an irreducible ceiling on end-to-end RAG quality in those regions.
>
> We note that while RAG motivates the importance of retrieval quality, the contribution of this paper is a general framework for information retrieval evaluation. The semantic stratification methodology applies to any retrieval system and evaluation metric, independent of whether a generation component is present.

---

> > ### Author Rebuttal · Reviewer_VJKp · 2026-04-03
> >
> > Thanks for your rebuttal. I will raise my score.

---

### Official Review · Reviewer_snb4 · 2026-03-13

**Soundness:** 3
**Presentation:** 3
**Significance:** 3
**Originality:** 2
**Overall Recommendation:** 4
**Confidence:** 3

**Summary:**

The paper studies whether current retrieval benchmarks adequately evaluate retrieval systems. It argues that existing evaluation query sets often fail to cover large portions of the semantic space of the document corpus, which can bias aggregate metrics. To address this, the authors propose a semantic stratification framework that organizes documents into entity-based clusters and constructs evaluation queries to ensure coverage across semantic and structural regimes.

**Compliance With Llm Reviewing Policy:**

Affirmed.

**Final Justification:**

My concerns are resolved. I think current score is reasonable for this paper.

**Key Questions For Authors:**

The framework depends on LLM-generated entities and queries. How sensitive are the results to the choice of model or prompt used for entity extraction and query generation?

How does the proposed evaluation protocol compare to human-curated query sets in terms of realism and reliability?

The proposed framework assumes that semantic clusters derived from entity graphs correspond to meaningful retrieval regimes. Did the authors evaluate the stability of these clusters across different random seeds or alternative clustering methods?

If a practitioner wants to apply this framework, what is the approximate computational cost of the full pipeline (entity extraction, graph construction, clustering, query generation)? Some estimates would help understand its practical usability.

**Limitations:**

Yes

**Strengths And Weaknesses:**

The formulation of evaluation as a stratified estimation problem is clear, and the analysis connecting regime coverage to bias in aggregate metrics is well motivated. The empirical study across multiple datasets and retrieval paradigms also helps support the argument that retrieval performance can vary significantly across semantic regions.

However, some of the pipelines rely heavily on LLM-generated entities and queries, which introduces uncertainty about the reliability of the evaluation process itself. The paper does not fully analyze how sensitive the results are to these choices.

The paper is clearly written and easy to follow. The motivation, framework, and experimental sections are structured well, and figures help illustrate the idea of semantic coverag

The idea of analyzing evaluation coverage using corpus-level structure is interesting, and framing retrieval evaluation as a stratified estimation problem is a reasonable conceptual contribution.

---

> ### Author Rebuttal · Authors · 2026-03-31
>
> We thank the reviewer for their thoughtful questions and we hope that our added analysis will strengthen the robustness of our arguments.
>
> **[W] LLM pipeline reliability & [Q1] Sensitivity to model/prompt choice**
>
> We agree that LLM dependence warrants scrutiny. We will respond through both direct argument and an additional robustness ablation study below.
>
> We address each component separately. The extracted entities are predominantly well-known concepts (e.g., "logistic regression," "cardiovascular disease"), domain-specific terminology (e.g., "Cox proportional hazards"), and named entities (e.g., "GlaxoSmithKline"). Modern LLMs show strong agreement on this task. Our normalization pipeline (Appendix A.1) further reduces surface-level variation between models. Importantly, our resolution ablation (see Q3 below) demonstrates that conclusions are robust to a 4x change in cluster count (155-669 clusters), suggesting that moderate perturbations in the entity set would produce smaller structural shifts than this and are unlikely to change diagnostic conclusions. We acknowledge that a systematic multi-model extraction comparison is valuable future work.
>
> The generation component (Sec. 4.3) serves to demonstrate that coverage gaps can be systematically addressed. While generated queries are inherently imperfect, the synthetic queries sufficiently exercise previously untested semantic regions, as evidenced by the coverage improvements in Table 4. We also enforce diversity in query length and specificity to better match real-world search patterns (details in Appendix C). A more detailed study of query generation quality is a natural direction for future work.
>
> **[Q2] Comparison to human-curated query sets**
>
> The evaluation protocol is independent of how queries are curated. It applies equally to human-written, LLM-generated, or community gathered query sets. In fact, the core analysis in this paper (Tables 2-4, Figures 3-5) is performed entirely on the original BEIR queries. The framework's role is not to replace human curation but to diagnose coverage gaps in any existing query set and guide where additional curation effort is most needed.
>
> **[Q3] Cluster stability across seeds and clustering methods**
>
> We conducted a resolution sensitivity analysis on NFCorpus across $\gamma \in$ {10, 15, 25, 40, 60}:
>
> | $\gamma$ | Clusters | sigma_within | $\Delta$ VRR | J VRR | Cluster nDCG (min / max) |
> |-------|----------|-------------|-----------|-------|--------------------------|
> | 10 | 155 | 0.310 | 0.104 | 0.219 | 0.436 / 0.632 |
> | 15 | 210 | 0.305 | 0.131 | 0.228 | 0.438 / 0.643 |
> | 25 | 313 | 0.306 | 0.125 | 0.237 | 0.432 / 0.739 |
> | 40 | 474 | 0.305 | 0.133 | 0.286 | 0.435 / 0.856 |
> | 60 | 669 | 0.304 | 0.137 | 0.326 | 0.401 / 0.856 |
>
> Across this 4x range in cluster count, all core findings hold. Within-stratum homogeneity is consistent. The variance reduction ratios (VRR) for dispersion and Jaccard — measuring how well our structural signals predict nDCG@10 and Recall@100 respectively — remain stable. At finer granularities, the min/max spread in per-cluster nDCG widens (0.40–0.86 at $\gamma=60$), revealing increasingly differentiated performance across semantic regions.
>
> Qualitatively, clusters are coherent at every level. The difference is granularity, not quality. At $\gamma=10$, a single cluster covers all statistical modeling (285 entities); at $\gamma=25$ this splits into survival analysis, meta-analysis, and significance testing; at $\gamma=60$, further refinement separates Cox regression from age-adjusted models.
>
> $\gamma$ is not a hyperparameter that needs optimization, it is a granularity dial that practitioners tune to their diagnostic needs. A broad coverage audit might employ a lower $\gamma$ to check whether major corpus regions are tested, while a targeted failure analysis might use a higher $\gamma$ to isolate specific subdomains. Since the entity extraction and embedding are computed once and cached, re-running clustering at a new $\gamma$ is fast, making it practical to explore multiple granularities. As practical guidance, we recommend validating that: (1) clusters are interpretable (spot-check 10-20 cluster descriptions), (2) no single cluster dominates >20% of entities, and (3) <5% of entities are isolated. These checks converge quickly; our per-dataset tuning required at most 3 trials. We have added this analysis and practitioner guidance to the revised paper.
>
> **[Q4] Computational cost**
>
> Entity extraction for small datasets (<100k documents) is upper-bounded by $\textdollar10$, generated with gpt-4o-mini. For larger datasets such as MS MARCO or HotpotQA (5M+ documents), costs rise to the hundreds of dollars due to a higher number of documents. Clustering entities comes at no token cost. Query generation costs less than five cents per query. These costs are a one-time investment. Once the semantic structure is constructed, it can be reused without additional costs.

---

> > ### Author Rebuttal · Reviewer_snb4 · 2026-04-02
> >
> > My concerns are resolved. I think current score is reasonable for this paper.

---

### Decision · Program_Chairs · 2026-04-30

**Decision:**

Accept (regular)

**Comment:**

All reviewers agreed on the importance of rethinking RAG evaluation, supported by empirical evidence. While many concerns were raised during the rebuttal, all major concerns are addressed so the reviewers are voted for acceptance.